# Structural basis of host recognition and biofilm formation by *Salmonella* Saf pili

**Longhui Zeng[†], Li Zhang[†], Pengran Wang[†], Guoyu Meng***

State Key Laboratory of Medical Genomics, Shanghai Institute of Hematology, Rui-Jin Hospital, Shanghai JiaoTong University School of Medicine and School of Life Sciences and Biotechnology, Shanghai JiaoTong University, Shanghai, China

**Abstract** Pili are critical in host recognition, colonization and biofilm formation during bacterial infection. Here, we report the crystal structures of SafD-*dsc* and SafD-SafA-SafA (SafDAA-*dsc*) in Saf pili. Cell adherence assays show that SafD and SafA are both required for host recognition, suggesting a poly-adhesive mechanism for Saf pili. Moreover, the SafDAA-*dsc* structure, as well as SAXS characterization, reveals an unexpected inter-molecular oligomerization, prompting the investigation of Saf-driven self-association in biofilm formation. The bead/cell aggregation and biofilm formation assays are used to demonstrate the novel function of Saf pili. Structure-based mutants targeting the inter-molecular hydrogen bonds and complementary architecture/surfaces in SafDAA-*dsc* dimers significantly impaired the Saf self-association activity and biofilm formation. In summary, our results identify two novel functions of Saf pili: the poly-adhesive and self-associating activities. More importantly, Saf-Saf structures and functional characterizations help to define a pili-mediated inter-cellular oligomerizaiton mechanism for bacterial aggregation, colonization and ultimate biofilm formation.

DOI: https://doi.org/10.7554/eLife.28619.001

**\*For correspondence:**
guoyumeng@shsmu.edu.cn

[†]These authors contributed equally to this work

**Competing interests:** The authors declare that no competing interests exist.

## Introduction

Biofilms are communities of microorganism embedded in a complex extracellular polymeric substance (EPS) matrix (*Costerton et al., 1995*). Bacterial cells in a biofilm can become more resistant to most antimicrobial agents and host defenses than their planktonic counterparts (*Burmølle et al., 2010*; *Mah and O'Toole, 2001*; *Prosser et al., 1987*). Biofilm development is a dynamic progress, and it can be mainly described in three stages including attachment, maturation and dispersion (*Rabin et al., 2015*). For bacterial pathogenesis, the pili-mediated attachment to the mucosal epithelial layers in human is often thought to be the first step to enable initial contact, to form a microcolony, to carry out invasion and to evade the host immune system that lead to acute and chronicle infections (*Proft and Baker, 2009*; *Sauer et al., 2000*; *Zavialov et al., 2007*). In addition, it has been reported that bacterial pili can participate in cell-cell interaction, leading to bacterial aggregation and micro-colony formation (*Lo et al., 2013*; *Wright et al., 2007*). As demonstrated in *Salmonella enterica* serovar Typhimurium, type I pili are necessary to establish initial attachment with the epithelium, while at least three other types of fimbriae (Lpf, Pef and Tafi) are required for cell-cell interaction, micro-colony growth and biofilm maturation (*Ledeboer et al., 2006*).

*Salmonella enterica*, a Gram-negative and food-borne enteric pathogen, is a common cause of human and animal abdominal complications like typhoid fever and gastroenteritis (*Silva et al., 2014*). It can be subdivided into seven subspecies, designated as I, II, IIIa, IIIb, IV, VI and VII (*Reeves et al., 1989*). Over 99% clinical isolates belong to *Salmonella enterica* subspecies I, which account for most *Salmonella* infections in humans and animals (*Folkesson et al., 1999*). Most serotypes of *Salmonella enterica* are able to form biofilms, which significantly increase their survival in a

variety of environments and hosts, and enhance resistance to multiple antimicrobials (*Papavasileiou et al., 2010*; *Sheffield et al., 2009*).

Saf pili are often found in clinical isolates of *Salmonella enterica* (*Folkesson et al., 1999*; *Townsend et al., 2001*). The *saf* operon is located on *Salmonella enterica* centisome seven genomic island (*Folkesson et al., 2002*), which consists of four contiguous genes encoding the major subunit (SafA), periplasmic chaperone (SafB), outer membrane usher (SafC) and minor subunit (SafD) (*Figure 1A*). Saf pili are assembled by the chaperone-usher (CU) secretion pathway (*Remaut et al., 2006*; *Zavialov et al., 2007*). The nascent SafA and SafD subunits are transported from the cytoplasm into the periplasm via the SEC machinery (*Stathopoulos et al., 2000*). The periplasmic chaperone, SafB, donates its G1 strand to complete the correct folding of SafA or SafD. Hence, this process is also known as donor strand complementation (DSC) (*Choudhury et al., 1999*; *Sauer et al., 1999*). The subunit-chaperone binary complexes migrate to the outer membrane usher, where subunits polymerize into a linear filament via a mechanism known as donor strand exchange

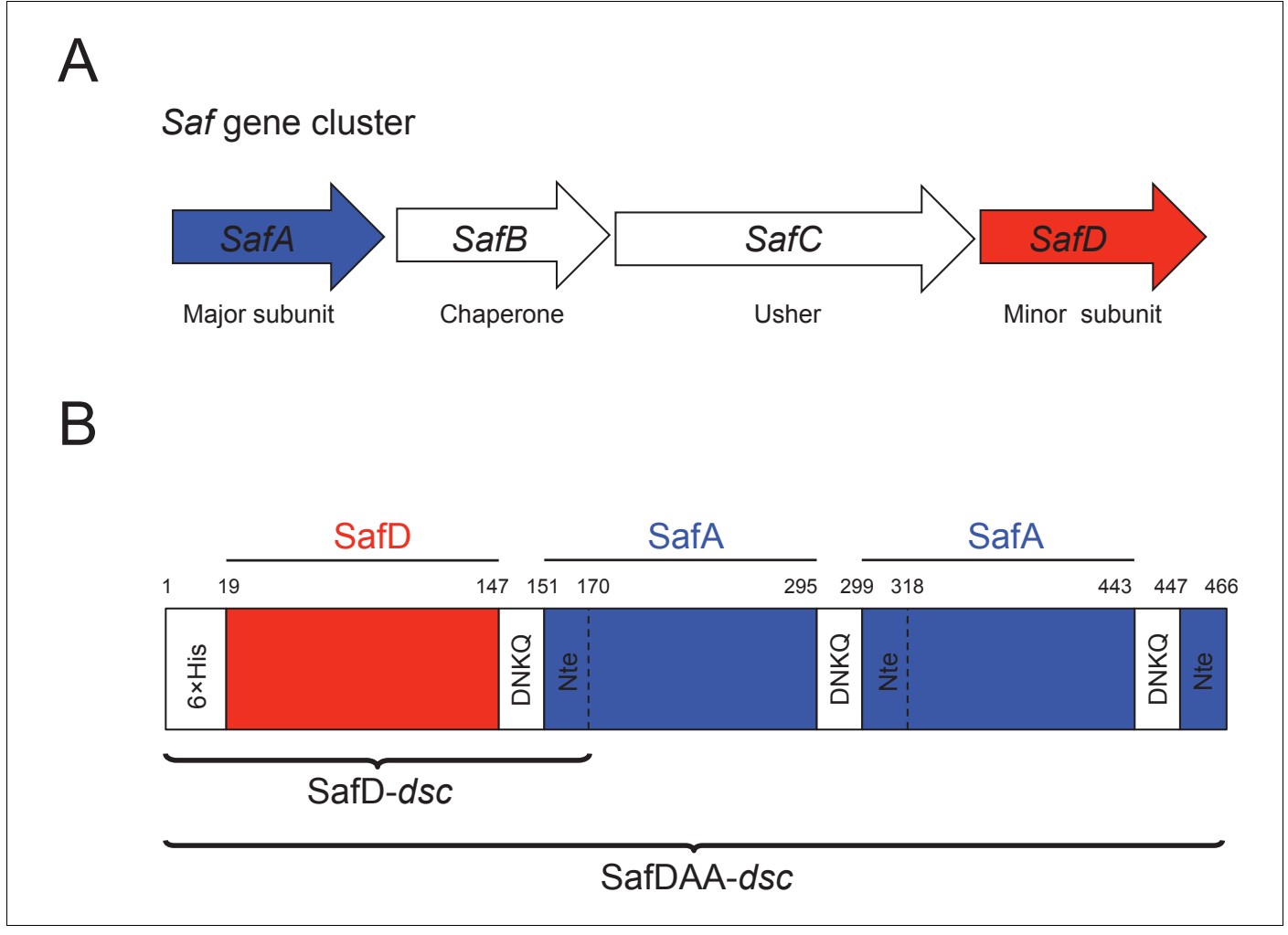

**Figure 1.** Recombinant expression strategy of SafD-*dsc* and SafDAA-*dsc*. (A) *SafABCD* gene cluster in *Salmonella enterica*. SafA, the major subunit. SafB, the periplasmic chaperone, SafC, the outer membrane usher, and SafD, the minor subunit. (B) Schematic diagram of recombinant SafD-*dsc* and SafDAA-*dsc*. The artificial linker, enabling the linkage of subunits, is composed of four residues DNKQ. A 19-residue donor strand (termed Nte from SafA) is used to ensure the correct folding of pilin subunits, allowing the stable expression of SafD-*dsc* and SafDAA-*dsc* in *E.coli*.

DOI: https://doi.org/10.7554/eLife.28619.002

The following figure supplement is available for figure 1:

**Figure supplement 1.** Sequence alignment of SafD (**A**) and SafA (**B**) in *Salmonella enterica* strains.

DOI: https://doi.org/10.7554/eLife.28619.003

(DSE) (*Sauer et al., 2002*; *Zavialov et al., 2003*). In the DSE step of Saf pili assembly, the N-terminal extension (Nte) of an incoming SafA replaces the chaperone G1 strand (in SafB) via a zip-in-zip-out mechanism (*Remaut et al., 2006*; *Rose et al., 2008*), giving rise to a remarkable polymer of SafD-(SafA)$_n$ (n > 100).

Concerning the structures in Saf pili, SafA subunit and SafB chaperone have been characterized by X-ray crystallography and EM single particle analysis (*Remaut et al., 2006*; *Salih et al., 2008*). SafD, a putative adhesin/invasin, was demonstrated to have vaccination potential in mice model (*Strindelius et al., 2004*). The structure of SafD has not been reported. As for biological functions of Saf pili, previous studies showed that Saf pili are required for *Salmonella* infection against pig (*Carnell et al., 2007*). However, genetic knockout of *SafA* in *S. typhimurium* showed little difference in mouse virulence when compared to wild-type *Salmonella* strains (*Folkesson et al., 1999*). In this study, we aim to provide more structural and functional insights into this important pilus subtype. By X-ray crystallography, we firstly determined the crystal structures of SafD-*dsc* and SafDAA-*dsc* to the resolutions of 2.2 and 2.8 Å, respectively. The SafDAA-*dsc* structure and its unexpected oligomerization verified by SAXS analysis led to the investigation and discovery of the novel biofilm formation activity of Saf pili, supported by bead/cell aggregation and biofilm formation assays. Furthermore, the structure-based mutagenesis targeting the SafDAA-SafDAA dimers consistently disrupted SafDAA-*dsc* oligomerization activity and Saf-driven bacterial aggregation. All these have led to the proposal of a novel self-associating pili subclass, in which the flexible pili (from different bacteria) might interact/intertwine with each other in trans to enable micro-colony formation, aggregation, and further biofilm development.

## Results

### Structure of SafD-dsc

In order to obtain soluble SafD, we engineered a self-complemented SafD-*dsc*, in which the N-terminal extension of SafA (i.e. the G strand) is fused with the C-terminus of SafD via an artificial DNKQ linker (*Figure 1B*). The crystal structure of SafD-*dsc* was determined by molecular replacement using AfaD (PDB code: 2IXQ) as search model. The final model was refined to 2.2 Å resolution using program PHENIX.REFINE (*Adams et al., 2010*) (*Table 1*). The SafD-*dsc* structure reveals a classic immunoglobulin-like fold, in which A-G strands form a β-sandwich fold with two β sheets packed against each other (*Figure 2AB*). Based on sequence alignment (*Figure 2E* and *Figure 1—figure supplement 1A*), SafD is thought to be a putative adhesin/invasin. Using Dali server (*Holm and Rosenström, 2010*), SafD is predicted to belong to the Afa/Dr poly-adhesin subfamily containing DraD (r. m.s.d. of 1.5 Å for 134 Cα atoms), HdaB (1.6 Å for 143 Cα atoms), AfaD (2.5 Å for 142 Cα atoms), AafB (1.1 Å for 129 Cα atoms) and AggB (1.5 Å for 133 Cα atoms) (*Figure 2—figure supplement 1A*). In previous studies, it has been demonstrated that AfaD adhesin/invasin, which shares 30% sequence homology with SafD, can recognize host recetor integrin α5/β1 and α5/β3 (*Cota et al., 2006*; *De Greve et al., 2007*). Interestingly, SafD-*dsc* structure harbors one highly negative charged patch that could constitute the binding site of the unknown host receptor (*Figure 2C*). Despite initial attempt of single-point site-directed mutagenesis failed to verify this putative binding site (data not shown), more characterizations with candidate host receptor should be carried out for further understanding.

Similar to the published SafA-A$_{Nte}$ structure (*Remaut et al., 2006*), the P1-P5 residues (i.e. Gln9, Lys11, Val13, Ile15 and Phe17) of donor G strand interacts directly with the hydrophobic groove in SafD (*Figure 2A*). However, in P* position (a favored interaction that is thought to stabilize the exchanged product during DSE (*Rose et al., 2008*), Phe3 is no longer in direct interaction with SafD. Instead, the sidechain of Leu4 could interact with the hydrophobic core of SafD (termed P** procket), mimicking the capping role of Phe3 (*Figure 2A* and *Figure 2—figure supplement 1B*). Consistently, the bulky hydrophobic residues in the positions 3 and 4 are conserved in SafA (*Figure 1—figure supplement 1B*).

Another notable feature of SafD-*dsc* lies in the P5 pocket (*Figure 2D*). Previous studies have shown that the local enviroment around P5 is important for DSE (*Remaut et al., 2006*; *Verger et al., 2006*). The asymmetric unit (ASU) contains two SafD-*dsc* molecules. The superimposition of the SafD molecules shows interesting structural variations/flexibilities in Loop$_{A-B}$ on top of the

**Table 1.** Data collection and structure refinement statistics of SafD-dsc and SafDAA-dsc.

**Data collection**

| Protein | SafD-dsc | SafDAA-dsc |
|---|---|---|
| Space group | $P2_12_12_1$ | $C2$ |
| Unit cell dimension (A) | | |
| a | 32.5 | 133.3 |
| b | 49.7 | 66.1 |
| c | 148.8 | 187.7 |
| β (°) | | 96.2 |
| Molecule per ASU | 2 | 3 |
| Derivative | NativeNative | |
| Source/Station* | BL17U | BL17U |
| Wavelength (Å) | 0.979i | 0.9793 |
| Resolution range (Å) | 74.4 - 2.2 | 93.3 - 2.8 |
| Observations ($1/s(1) > 0$) | 48909 230250 | |
| Unique reflections ($1/s(1) > 0$) | 11971 (1640) | 38710 (5615) |
| High-resolution shell (Å) | 2.32 - 2.20 | 2.95 - 2.80 |
| Rsym (%)[†,c]: | 18.5 (67.9) | 14.0 (148.7) |
| $<I/s(I)>$[‡]: | 6.8 (2.2) | 8.1 (1.3) |
| Completeness[‡] (%): | 92.9 (88.5) | 96.0 (96.1) |
| Redundancy[‡]: | 4.1 (4.3) | 5.9 (6.0) |
| $CC_{1/2}$ | 0.98 (0.52) | 0.99 (0.44) |
| **Structure refinement** | | |
| Resolution range (Å) | 74.4 - 2.2 | 93.3 - 2.8 |
| R-factor (%) | 19.9 | 21.8 |
| R-factor (high resolution shell)[§] | 25.7 | 36.3 |
| Rfree (%)[#] | 23.3 | 25.5 |
| Rfree (high-resolution shell) | 33.6 | 39.5 |
| Total number of non-hydrogen atoms | 2254 | 9593 |
| Protein atoms | 2129 | 9543 |
| Water molecules | 125 | 50 |
| R.m.s. deviations:[¶] | | |
| Bond length (Å) | 0.003 | 0.006 |
| Bond angle (°) | 0.666 | 0.993 |
| Main chain B-factors (Å²) | 1.665 | 3.533 |
| Side chain B-factors (Å²) | 4.378 | 9.652 |
| Wilson B-factor (Å²) | 19.7 | 66.9 |
| Average B-factor protein atoms (Å²) | 26.2 | 93.2 |
| Ramachandran statistics (%) | | |
| Most favored region | 98.5 | 95.6 |
| Allowed regions | 1.1 | 4.1 |
| Outliers | 0.4 | 0.3 |

*Beamline designations refer to the Shanghai Synchrotron Radiation Facility, Shanghai, P. R. of China.
[†]$R_{sym}=S(I-<I>)^2/SI^2$.
[‡]overall, high resolution shell in parentheses.
[§]high resolution shell: 2.370–2.200 Å (SafD-*dsc*) and 2.870–2.800 Å (SafDAA-*dsc*).
[#]$R_{free}$ calculated using 5% of total reflections omitted from refinement.

[1]R.m.s. deviations report root mean square deviations from ideal bond lengths/angles and of *B*-factors between bonded atoms (*Engh and Huber, 1991*).
DOI: https://doi.org/10.7554/eLife.28619.006

P5 pocket (*Figure 2D*). In line with this observation, the flexibility of Loop$_{A-B}$ is also frequently observed in other pilin-Nte complexes (*Figure 2—figure supplement 1A*). Supportively, molecular dynamic (MD) simulation of *E.coli* pilin-Nte complex suggested that this loop might play a regulatory role in DSE (*Ford et al., 2012*).

## Structure of SafDAA-dsc and the poly-adhesive activity

In order to obtain the atomic detail of Saf pilus, we engineered a recombinant ternary complex, Saf-DAA-*dsc*, to mimic the pilus tip (*Figure 1B*). The structure of SafDAA-*dsc* was determined to 2.8 Å resolution by X-ray crystallography. The SafD, SafA1, SafA2 and SafA3$_{Nte}$ have assembled into a thread-like shape with 26 Å in diameter and 136 Å in length/height (*Figure 3* and *Figure 3—figure supplement 1A*). In ASU, three SafDAA-*dsc* molecules with an extended 'I'-like or a curvy 'L'-like architecture can be observed (*Figure 3—figure supplement 1A*). In the 'L'-architecuture, SafA2 swings ~26 Å upward with its central axis in 90° angle with the rest of the structure. As a result, the overall height of 'L'-like SafDAA-dsc (~110 Å) is significantly shorter.

Structural comparisons among SafDAA-*dsc* structures show that: (1) The overall Ig-fold of SafD and SafA subunits is strictly conserved; (2) The relative orientation between SafD and SafA1 is also conserved. The r.s.m.d. values between different SafD-SafA1 molecules range from 0.5 to 0.8 Å for 290 Cα atoms. The intra-molecular interactions between SafD and SafA1 are mainly mediated by hydrogen bonds formed between Thr111, Phe14, Ala16 of SafD, and Gln143, Ser18, Asn6 of SafA1 (*Figure 3—figure supplement 1B*); (3) In marked contrast, there is no hydrogen bonding, electrostatic or hydrophobic interactions in between SafA1-SafA2 (*Figure 3—figure supplement 1A*). The SafA2 appears to be quite isolated and has the ability to move significantly to engage inter-molecular interaction as observed in the crystal. Consistently, as demonstrated by EM single-particle analysis, SafA-SafA is indeed flexible and different SafA-SafA orientations can be observed (*Salih et al., 2008*). (4) The isomerization and torsion angles of Pro20 are important for the overall architecture and the flexibility of SafDAA-*dsc*. It has been reported that the isomerization of proline residue is important for protein folding and diversity (*Lu et al., 2003*; *Nicholson and De, 2011*). This is also the case in pili assembly (*Li et al., 2009*). In between SafD and SafA1, Pro20 adopts a *cis*-configuration, enabling a kink in the subunit-subunit linker loop that, in turn, allows the formation of intramolecular hydrogen bonds (*Figure 3—figure supplement 1B*). This appears to shape the SafD-SafA1 into a conserved orientation, in which the G strand in SafD forms a 120° angle with the central axis of SafA1. In comparison, Pro20 in between SafA1-SafA2 adopts a *trans* configuration, allowing SafA2 moving freely away from SafA1 as observed in 'I'- or 'L'-like architectures (*Figure 3—figure supplement 1A*).

Next we want to know whether and how the flexible, thread-like SafDAA-*dsc* mediates host recognition. It has been reported that Saf pili are required for intestinal colonization (*Carnell et al., 2007*). In order to check whether the recombinant SafDAA-*dsc* still had adhesive activity, cell adherence assay with porcine intestinal columnar epithelial IPEC-J2 cells was performed (*Figure 4A*). In this experiment, 3 µM His-SafDAA-*dsc* was added onto the plates coated with IPEC-J2 cells. The binding was monitored by anti-His antibody at different time points using ELISA approach. Compared to control (i.e. the His-SH3 protein derived from human c-SRC that is exclusively expressed in cytoplasm, and hence is a good reference for non-specific interaction against IPEC-J2 cells), the cell adherence assay showed that the recombinant SafDAA-*dsc* was functional and preserved the adhesive function of the Saf pilus (*Figure 4A*). Based on sequence and structural alignment, Saf pili are predicted to belong to a FGL pili subgroup, which is often associated with poly-adhesive activity (*Zavialov et al., 2007*). In order to check whether SafD and SafA are required for binding, SafDAA was progressively truncated into SafDA, SafAA, SafD and SafA, followed by cell adherence characterization. When SafAA, SafD or SafA was incubated with IPEC-J2 cells, very little binding was observed (*Figure 4B*). In marked contrast, SafDAA yielded a strong binding signal (*Figure 4B*). In addition, the removal of SafA subunit (SafDA) reduced host recognition. Altogether, these data have led to the proposal of a poly-adhesive activity for Saf pili, in which SafD and SafA subunits can bind

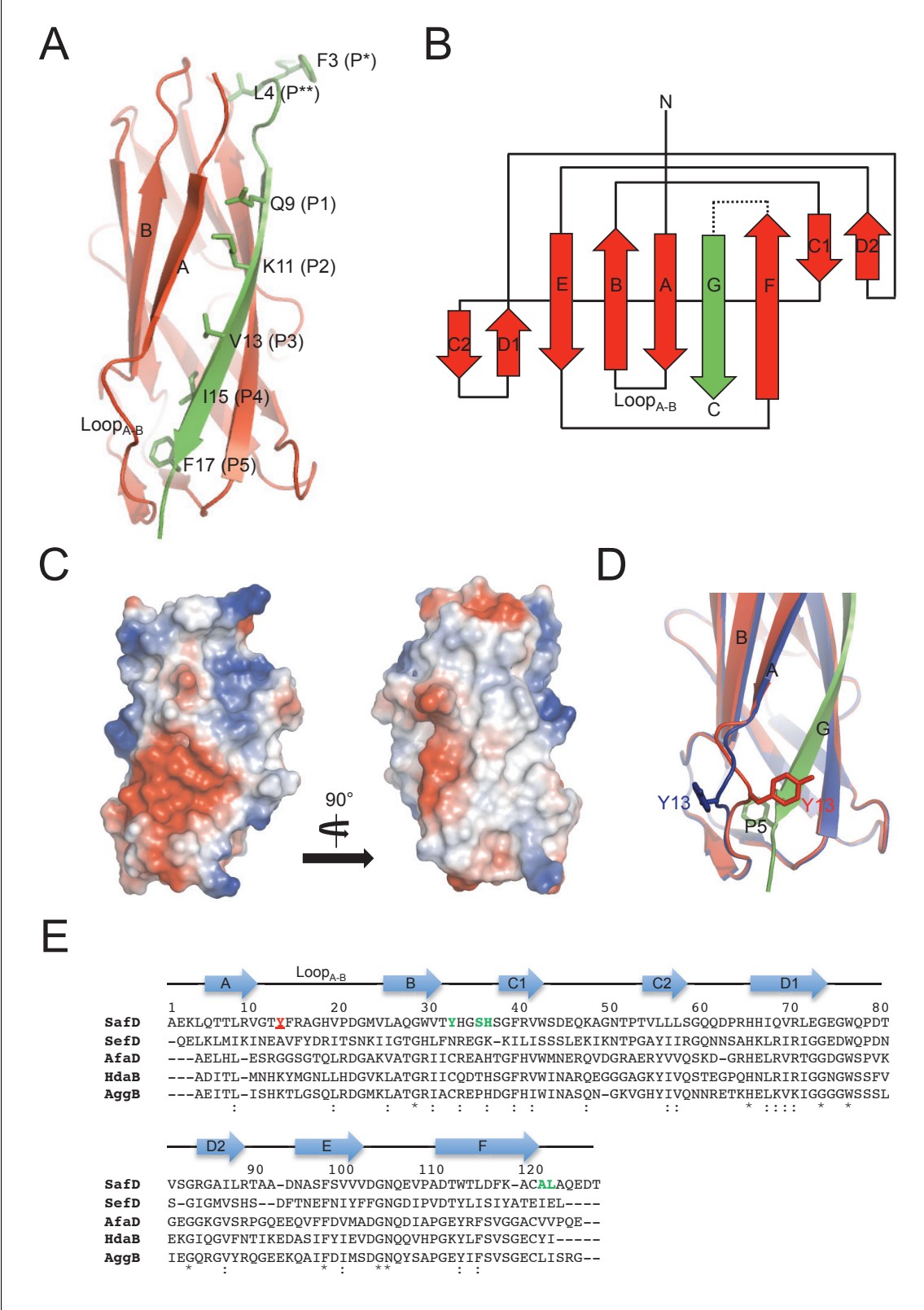

**Figure 2.** Crystal structure of SafD-*dsc*. (**A**) The cartoon representation of SafD-SafA_{Nte}. The SafD is colored in red and SafA_{Nte} in green. The residues, Leu4, Gln9, Lys11, Val13, Ile15, Phe17, corresponding the P** and P1-P5 pockets, respectively, are shown in stick representation. (**B**) The topology diagram of SafD-*dsc*. The secondary structure of SafD is colored in red. The complementing G strand is colored in green. The artificial linker DNKQ is highlighted with dotted line. (**C**) Electrostatic surface of SafD-*dsc*. The surface is colored with electrostatic potential (blue for positive charge and red for

*Figure 2 continued on next page*

*Figure 2 continued*

negative charge). (D) Structural superimposition of SafD-*dsc* molecules presented in this report. The residue Tyr13 in Loop$_{A-B}$ is highlighted and shown in stick representation. (E) Sequence alignment of SafD, SefD, AfaD, DraD and AggB adhesins. The β-strands of SafD are shown with arrows on top of the sequences. The highly and relatively conserved residues are indicated with '*' and ':', respectively. Tyr13 is underscored and colored in red. Residues delineating the P** pocket are shown in green.

DOI: https://doi.org/10.7554/eLife.28619.004

The following figure supplement is available for figure 2:

**Figure supplement 1.** Protein diversity in SafD-*dsc*.

DOI: https://doi.org/10.7554/eLife.28619.005

concertedly to the unknown host cell receptors to enable intimate host:bacterium interaction and colonization via a zip-in mechanism (*Figure 4C*).

## SafDAA-SafDAA self-association

It is not uncommon that bacteria can employ self-associated adhesins for initial cell-cell interaction, and subsequent microcolony and biofilm formation (*Garnett et al., 2012a*; *Garnett and Matthews, 2012*; *Heras et al., 2014*; *Klemm et al., 2006*; *Meng et al., 2011*). Until now, although pili are increasingly recognized as an important contributor in biofilm development, it is not yet clear how the thread-like structure chained with repetitive Ig-fold subunits can mediate cell-cell interaction. In this report, unexpectedly, the SafDAA-*dsc* structures reveal two types of self-associating oligomerization in the crystal (*Figure 5BC*). In Type I dimerization, SafD and SafA1 interact with each other in a head-to-tail configuration mainly via inter-molecular hydrogen bonding involving residues Arg9, Thr7, Asn94 from SafD, and Glu8, Gln10 from SafA (*Figures 5B* and *6A*). Consistently, when comparing SafA1 and different SafA2s (that do not form direct hydrogen bonds with SafD), a local side-chain reshuffle involving Glu8 could be observed in Type I dimerization (*Figure 3—figure supplement 1C*). The second SafA2s that are not involved in inter-molecular interaction 'float' freely away from the dimeric interface. Consistently, 'I'- and 'L'-like architecture are both observed in Type I dimer (*Figure 5B*). In Type II dimerization, the SafD, SafA1 and SafA2 subuits make extensive interactions with the neighbour SafDAA-*dsc* via a '6–9'-like self-complementary architecture/surface (*Figure 5C*). In particular, the last SafA2 subunits have shifted up significantly to interact with the incoming SafD molecules. As result, the morphologies of SafDAAs in this dimer are restricted to 'L'-like architecture (*Figure 5C*). In marked contrast to Type I dimer, Type II dimerization displays few hydrogen bonds. The SafDAA-SafDAA appears to engage each other via self-complementary sufaces controled by Pro20 (*Figure 5A–C*). The residues Pro20, as described before, could influence the overall architecutre/surface, and hence self-self interaction. Taken together, these two types of head-to-tail-like engagements could give rise to a remarkable SafDAA trimer (*Figure 5D*).

In order to verify the SafDAA-SafDAA oligomerization in solution, the biophysical technique small-angle-X-ray scattering (SAXS) was used (*Figure 5EF*). In this experiment, the purified SafDAA-*dsc* was subjected to X-ray scattering. The experimental data were then compared with SafDAA monomers/dimers/trimer derived from crystallography. As shown in *Figure 5E*, the perfect match in crystal fitting with Chi$^2$ value of 2.74 suggested the existence of SafDAA-*dsc* monomer (79.7%), dimer (2.1%) and trimer (18.2%) in solution (*Figure 5F*). This is further supported by size exclusion chromatography - multi-angle light scattering (SEC-MALS) analysis. Consistent with SAXS analysis, WT SafDAA-*dsc* and mutants behaved mainly as monomer in gel filtraton column (*Figure 6B*), resemblance of the polymerization/depolymerization of self-associating Hap-Hap, Ag43-Ag43 interactions (*Heras et al., 2014*; *Meng et al., 2011*). However, when comparing the oligomerization activities of these proteins, it is clear that WT SafDAA-*dsc* could undergo much higher oligomerizaiton (*Figure 6B*), whilst the self-associating activities were severely disrupted by the mutations of T7A/R9A/N94A, E8A/Q10A and P20A (*Figure 6B*).

All these observations have provoked us to investigate the role of Saf pili self-association activity in biofilm development. Firstly, in order to investigate the self-associating activity of SafDAA-*dsc*, we firstly performed the latex beads assay as described in our previous study of self-associating Hap adhesin (*Meng et al., 2011*). As shown in *Figure 5—figure supplement 1*, the beads coated with the recombinant SafDAA-*dsc* protein aggregated together to form clusters. In comparison, the beads coated with SafD-*dsc*, SafA-*dsc* and BSA protein remained much isolated. Secondly, a pASK-

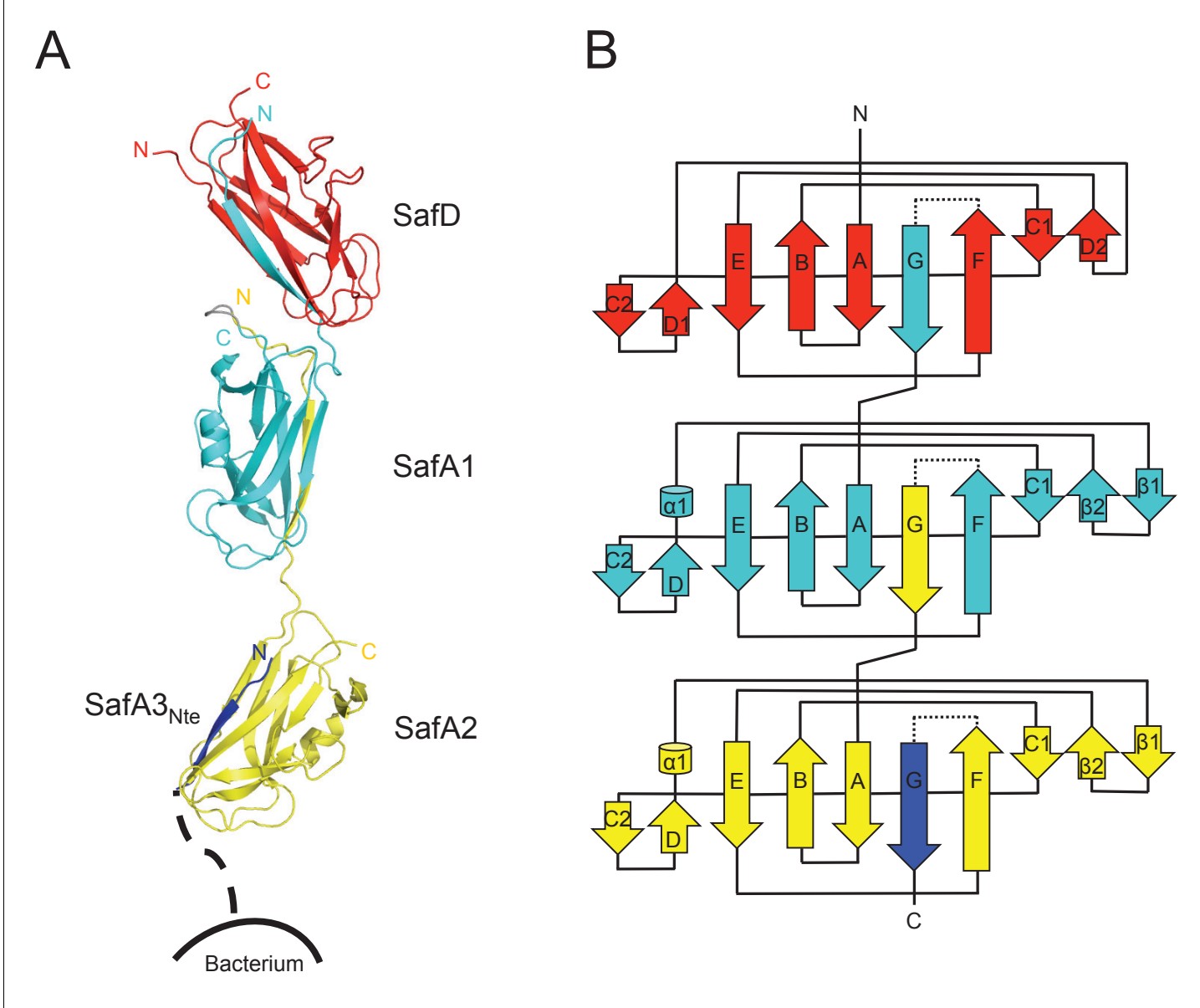

**Figure 3.** Crystal structure of SafDAA-*dsc*. (**A**) The cartoon representation of SafD-SafA1-SafA2-SafA3$_{Nte}$, colored in red, cyan, yellow and blue, respectively. N- and C-termini of each subunit are labelled. The other SafA subunits of Saf pili are shown with a dotted line. The artificial linker sequences DNKQ in between SafD-SafA1 is colored in gray. (**B**) The topology diagram of SafDAA-*dsc*. The D, A1, A2 and A3 subunits are colored using the same color scheme as described above.

DOI: https://doi.org/10.7554/eLife.28619.007

The following figure supplement is available for figure 3:

**Figure supplement 1.** The architectures of SafDAA-*dsc*.
DOI: https://doi.org/10.7554/eLife.28619.008

IBA4-Saf plasmid containing the intact *SafABCD* gene cluster was constructed and subjected to cell aggregation and biofilm formation assays (***Figure 7***). The expression and surfacing of Saf pili in *E. coli* were monitored by flow cytometry, immunoelectron microscopy and outer membrane protein extraction (***Figure 7ABF***). In agreement with beads aggregation observation, *E.coli* cells with WT Saf, but not ΔSafA, showed significant increase of cell-cell interaction and colony formation visualized by light microscopy (***Figure 7CD***). More importantly, when the perturbation/mutations were

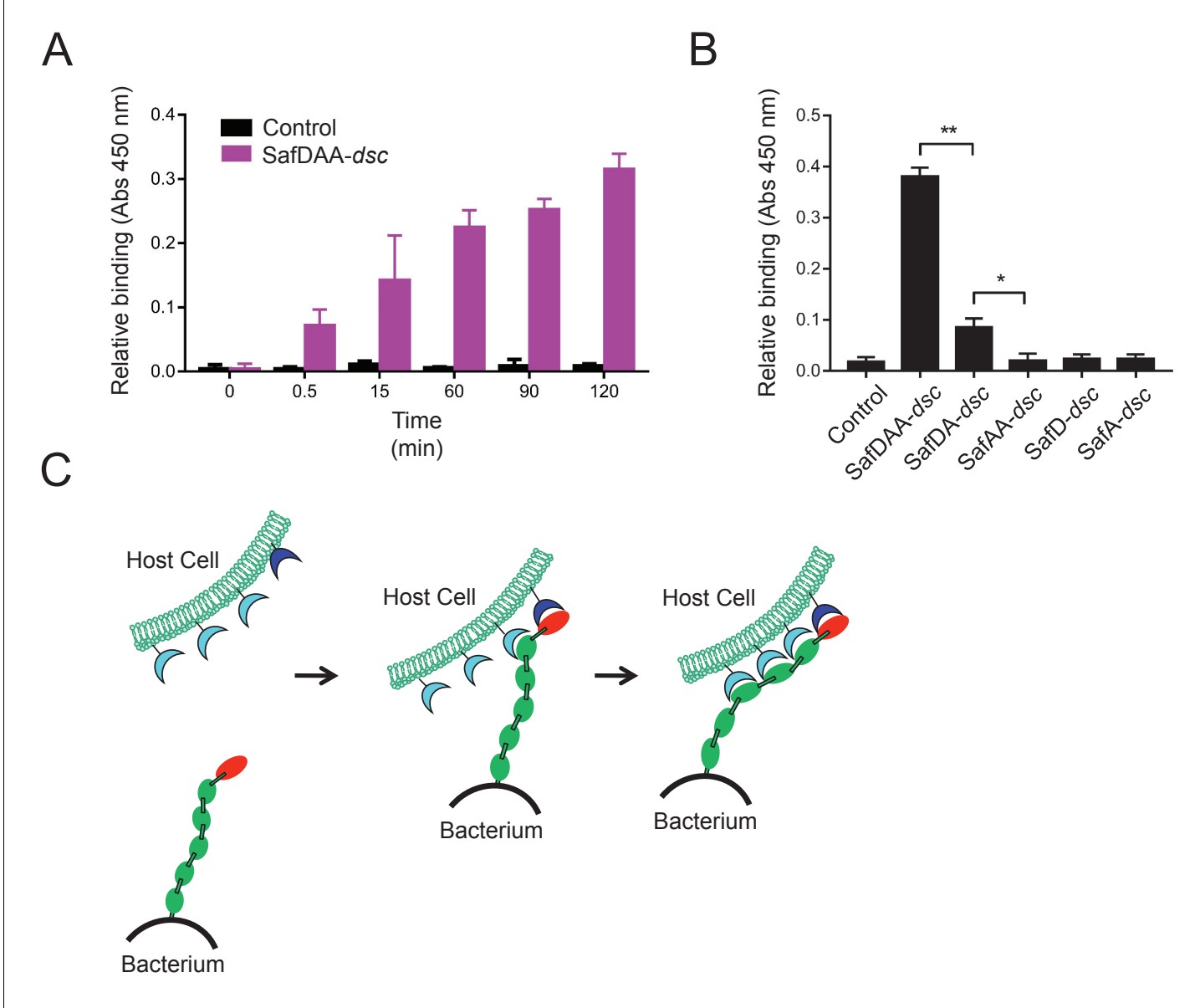

**Figure 4.** Host recognition by *Salmonella* Saf pili. (**A**) Cell adherence assay. 3 μM SafDAA-*dsc* was incubated with IPEC-J2 cells and the relative biniding of SafDAA-*dsc* was monitored at different time points up to 2 hr. 3 μM recombinant protein His-SH3 was used as control. (**B**) The binding activities of SafDAA-*dsc*, SafDA-*dsc*, SafAA-*dsc*, SafD-*dsc* and SafA-*dsc*. 10 μM recombinant Saf proteins were incubated with IPEC-J2 cells for 1 hr prior to ELISA analysis. All experiments have been done at least with three independent replicates. Values are means ±S.E. *p<0.05 and **p<0.01 (derived from student *t* test) are used to show statistically significant between recombinant derivatives. (**C**) The poly-adhesive host recognition mechanism by Saf pili. In the first step, the SafD (red) located in the distal tip of the pilus can mediate the initial host recognition. The unkonwn host receptor is colored in blue. In the second step, an intimate host:bacerium association leading to bacterial colonization and diseases can be formed by the sequential binding of SafA subunits (green) with a separate set of host receptors (cyan).

DOI: https://doi.org/10.7554/eLife.28619.009

The following source data is available for figure 4:

**Source data 1.** Source data for *Figure 4A*.
DOI: https://doi.org/10.7554/eLife.28619.010
**Source data 2.** Source data for *Figure 4B*.
DOI: https://doi.org/10.7554/eLife.28619.011

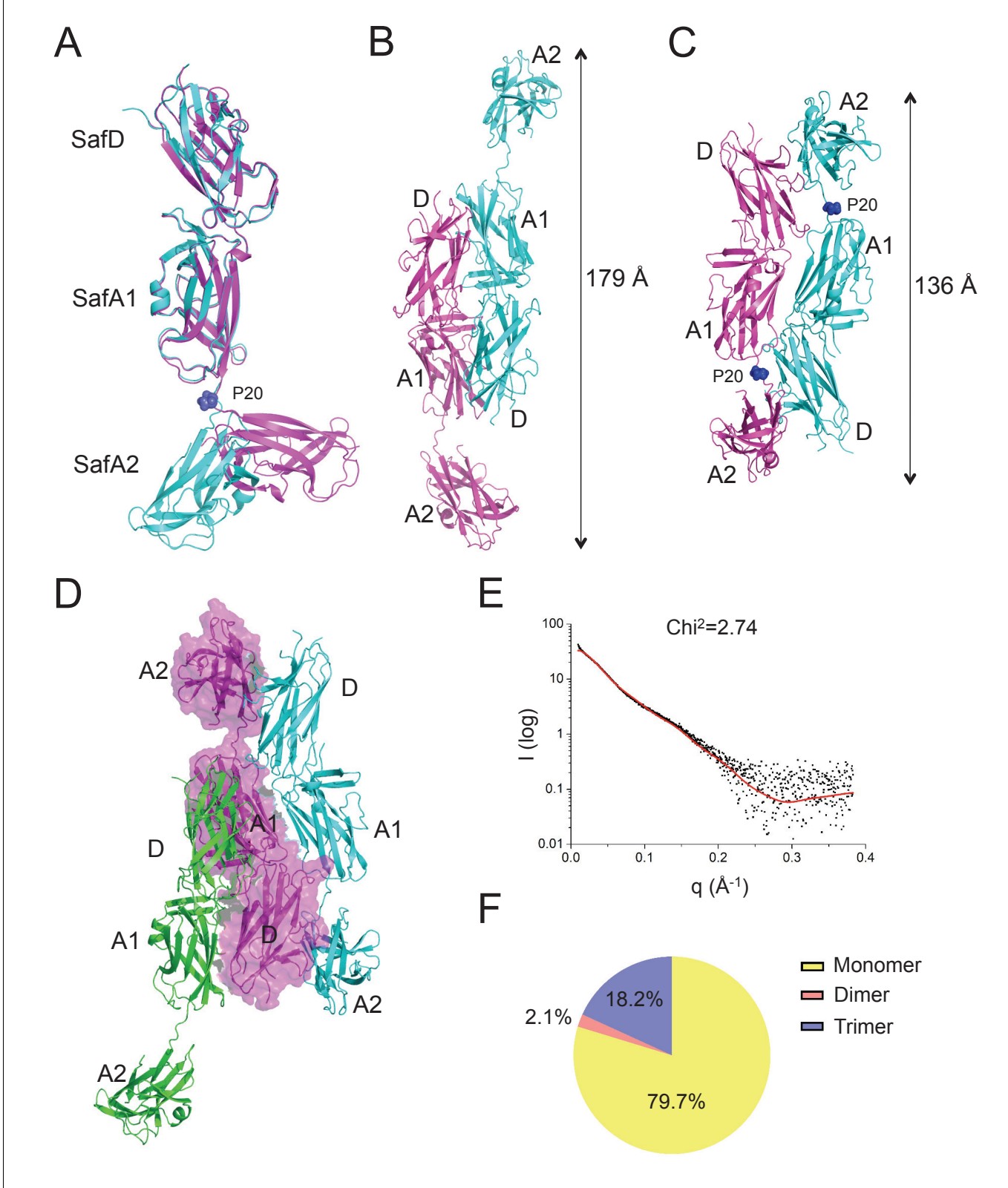

**Figure 5.** Self-associating SafDAA-SafDAA interaction. (**A**) The superimposition between 'I'- and 'L'-like SafDAA-*dsc*. The two molecules are colored in cyan and magenta, respectively. The Pro20 residue in the linker Loop$_{G-A}$ between SafA1 and SafA2 is highlighted with sphere representation and colored in blue. (**B**) The Type I dimerization. SafD-SafA1 (molecule 1, magenta) interacts with a nerighbouring counterpart (molecule 2, cyan) in a head-to-tail configuration. The overall length of the Type I dimer is 179 Å. (**C**) The Type II dimerization. Two SafDAA molecules intertwine with each other,

*Figure 5 continued on next page*

*Figure 5 continued*

enabling the intermolecular interaction between SafD (molecule 1, magenta) and SafA2 (molecule 2, cyan). As a result, the dimerization appears to give rise to a more intimate '6–9'-like self-association. The overall length of Type II dimer is 136 Å. (**D**) The SafDAA trimer. For the purpose of clarity, the SafDAA molecule in the middle is shown with transparent surface and the other two are shown in cartoon representations. (**E**) The SAXS characterization. Black line, experimental data. Red line, the theoretical scattering pattern derived from SafDAA multimers. (**F**) Oligomeric distribution estimated by SAXS analysis.

DOI: https://doi.org/10.7554/eLife.28619.012

The following source data and figure supplements are available for figure 5:

**Source data 1.** Source data for *Figure 5E*.
DOI: https://doi.org/10.7554/eLife.28619.015
**Figure supplement 1.** Visualization of the SafDAA-SafDAA self-association using the latex beads assay.
DOI: https://doi.org/10.7554/eLife.28619.013
**Figure supplement 1—source data 1.** Source data for *Figure 5 Figure 5* – figure supplementar 1B.
DOI: https://doi.org/10.7554/eLife.28619.014

introduced to the SafDAA-SafDAA dimeric interfaces or in the proline residue of Loop$_{G-A}$, the self-association suffered various degrees of disruption (*Figure 7CD*).

Similar results were observed in biofilm formation assay as monitord by crystal violet staining (*Figure 7E* and *Figure 7—figure supplement 1B*). Consistent with the results obtained in SEC-MALS and cell aggregation assay, doube and multiple mutations (T7A/R9A/N94A and E8A/Q10A) displayed an aggravated effect with increased disruption in Saf-mediated biofilm formation

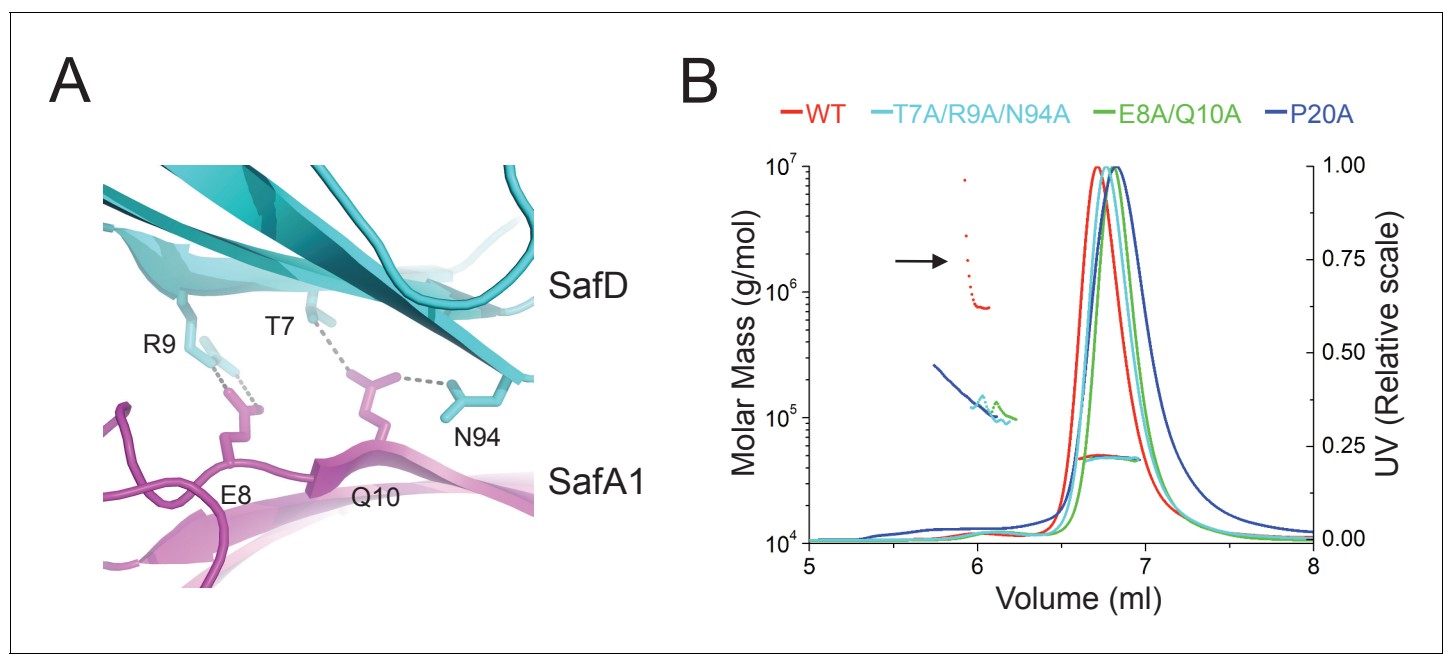

**Figure 6.** Structural determinants of SafDAA-SafDAA interaction. (**A**) The inter-molecular hydrogen bonding network in the dimeric interface of SafDAA-SafDAA. As shown in *Figure 5B*, the SafD-SafA1 intermolecular interaction is repetitive in Type I dimerization. For the purpose of illustration, one dimeric interface is shown. The SafD residues, Thr7, Arg9, Asn94 form four inter-molecular hydrogen bonds with the SafA residues, Glu8, Gln10. The residues are shown in stick representations, and the hydrogen bonds are highlighted with dash lines. (**B**) SEC-MALS characterization of SafDAA-*dsc* (red) and mutants (cyan, green and blue) in solution. The elution traces of molecular weight and UV are shown with dashed and solid lines, respective. The self-associating activity, monitored by higher order oligomerization, is highlighted with arrow. The structure-based mutations targeting the dimeric interface and Pro20 significantly impair the SafDAA oligomerization activity.

DOI: https://doi.org/10.7554/eLife.28619.016

The following source data is available for figure 6:

**Source data 1.** Source data for *Figure 6B*.
DOI: https://doi.org/10.7554/eLife.28619.017

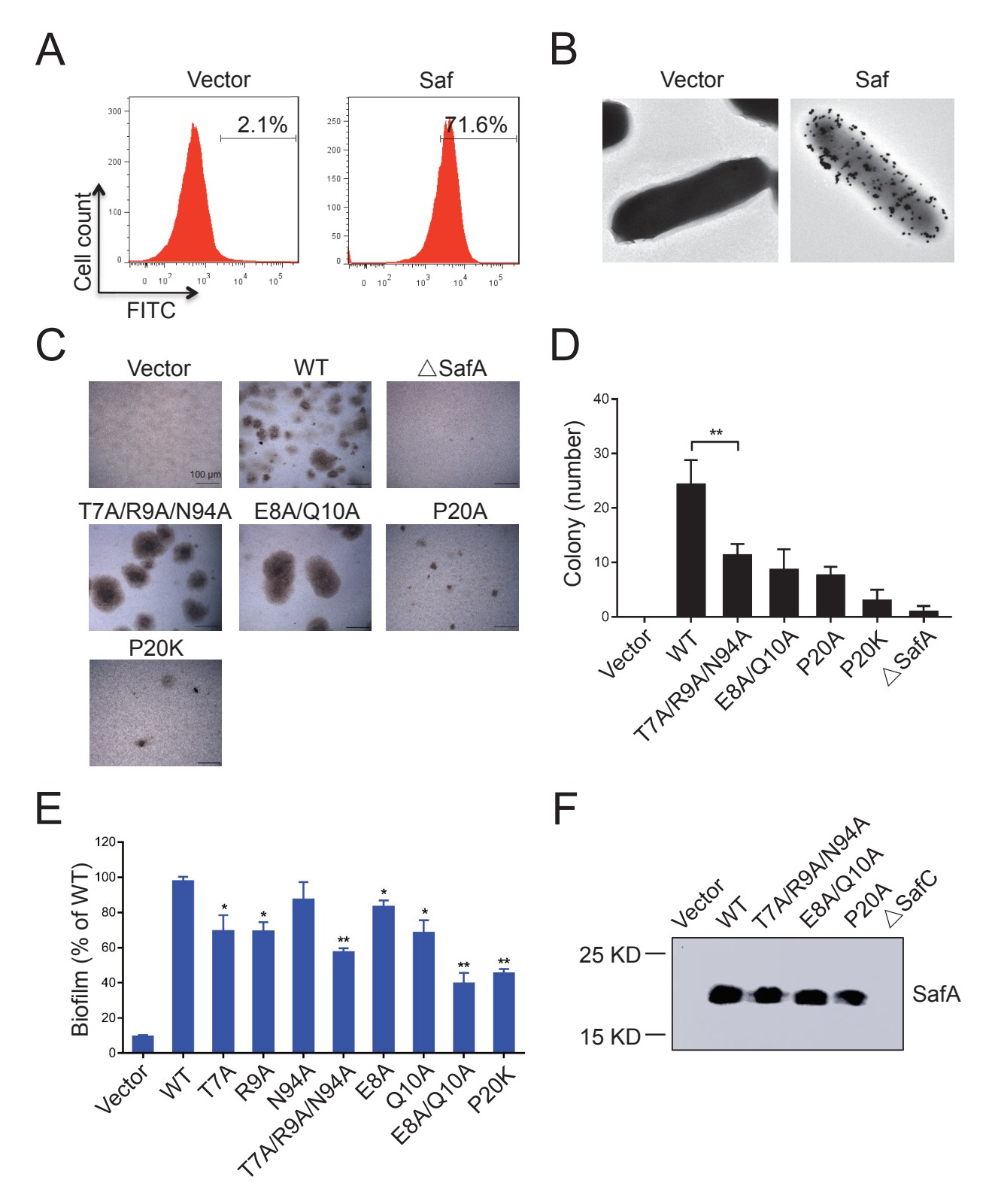

**Figure 7.** Inter-cellular SafDAA-SafDAA self-association drives biofilm formation. (**A**) The flow cytometry assay was used to monitor the expression and surfacing of Saf pili in LB medium. The quantified surfacing efficiency of cells harboring pASK-IBA4-Saf was 71.6%, in comparison to the background value of 2.1% for the empty pASK-IBA4 vector. (**B**) EM visualization of Saf surfacing using anti-strep immunogold. Top left: control cells that contained empty pASK-IBA4 vector. Top right, *E.coli* cells harboring pASK-IBA4-Saf. Both sets of the cells are incubated with anti-strep immunogod particles

*Figure 7 continued on next page*

*Figure 7 continued*

(black dots) prior to EM inspection. (C) Saf-driven cell aggregation visualized by light microscopy. After induction with 2% AHT for 24 hr in a 96-well plate, the colonization ability of WT Saf pili and muants were visualized by light microscopy. (D) Quantitative analysis of cell aggregation assay (C). The average number of colony formation for WT and mutants were calculated from three independent experiments in a 96-well plate. **p<0.01, statistically significant compared with WT. (E) Structure-based mutagenesis and biofilm formation assay. The wild-type (WT) Saf pili and the mutants that target the self-association were expressed in *E.coli* DH5α using pASK-IBA-Saf. The biofilm formation was quantified using crystal violet staining. For the purpose of comparison, the self-associating/biofilm-formation activities of mutants were normalized against WT and showed in percentage values. All experiments have been done at least with three independent replicates. Values are means ±S.E. from three independent experiments. *p<0.05 and **p<0.01 are used for statistic analysis between WT and mutants. (F) The expression levels of WT Saf pili and mutants in *E.coli*. The bacterial outer membrane (OM) fraction harboring WT Saf or mutants was extracted and subjected to western blot analysis using antibody against strep-SafA. The pASK-IBA4-SafABD, that is, ΔSafC, was used as negative control.

DOI: https://doi.org/10.7554/eLife.28619.018

The following source data and figure supplement are available for figure 7:

**Source data 1.** Source data for *Figure 7D*.
DOI: https://doi.org/10.7554/eLife.28619.020
**Source data 2.** Source data for *Figure 7E*.
DOI: https://doi.org/10.7554/eLife.28619.021
**Figure supplement 1.** Saf-driven bacterial aggregation.
DOI: https://doi.org/10.7554/eLife.28619.019

(*Figure 7E*). As for Type II Saf pili dimerization, P20K mutation, which was engineered to restrict the proline flexibility and to disrupt the complementary architecture/surface in Type II dimerization, also displayed disruptive effect on Saf-driven biofilm development (*Figure 7E*). This has led to the structural investigation of proline residue in different pili assemblies (*Figure 8*). The proline residue is frequently observed in the loop in between G- and A-strands, termed Loop$_{G-A}$. As shown in *Figure 8C*, the isomerization of Pro and the torsion angles of X-P-X (X for any amino acid) are indeed important structural determinants for the overall architectures and functions in CFA/I fimbriae (*Li et al., 2009*).

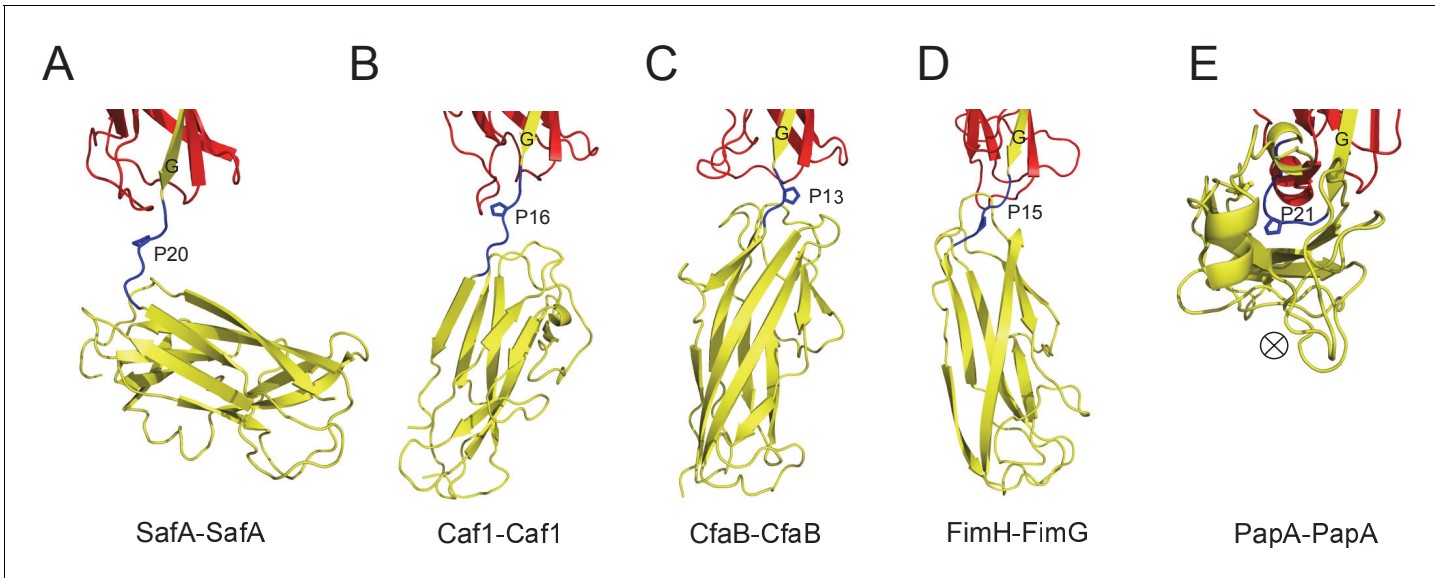

**Figure 8.** The Proline residue in Loop$_{G-A}$ controls the flexibility and overall architecture/shape of bacterial pilus. The structures of SafA-SafA in *Salmonella* atypical fimbriae (A), Caf1-Caf1 in Capsular F1 antigen pili (*Zavialov et al., 2003*) (B), CfaB-CfaB in Class 5 fimbriae (*Li et al., 2009*) (C), FimH-FimG in Type 1 pili (*Geibel et al., 2013*) (D), and PapA-PapA in P pili (*Hospenthal et al., 2016*) (E) are shown in cartoon representation. The preceding subunits are colored in red with the donor G strand in yellow. The following subunits are colored in yellow and the linker, Loop$_{G-A}$, are shown in blue with the proline residues highlighted with stick representation. In order to visualize the flexibility between subunits, all the structures are superimposed based on the coordinates of the preceding subunit (red). ⊗ indicates an inward Z direction of PapA.

DOI: https://doi.org/10.7554/eLife.28619.022

## Discussion

### Protein diversities in Saf pili

In Saf pili, in addition to the P1-P5 residues, Remaunt and colleages have demonstrated that the formation of SafA-SafA$_{Nte}$ is stabilized by an additional P* residue, Phe3 (*Remaut et al., 2006*). In SafD-*dsc*, a conserved Leu4 in SafA$_{Nte}$ was used to mimic Phe3, and the corresponding hydrophobic pocket is termed the P** pocket. The analysis with sequence alignment shows that Phe3 and Leu4, which are not conserved in other pilin subunits, are conserved among *Salmonella* SafAs (i.e. these two positions are often decorated with bulky hydrophobic residues) (*Figure 1—figure supplement 1B*). This seems to imply a unique structural signature in Saf pili. Further investigation should be carried out to test whether Phe3-Leu4 double safety capping in Saf pili assembly will affect its biological functions. Besides, another protein diversity lies in Tyr13 in SafD (*Figure 2D*). As previously characterized, the initial interaction between subunit and the incoming Nte peptide in P5 pocket is thought to be the starting point of DSE process (*Remaut et al., 2006*; *Verger et al., 2008*). MD simulation of in PapD-PapGII complex suggested a regulatory role of Loop$_{A-B}$. Based on these observations, it is reasonable to speculate that the Try13 sidechain of SafD might act as a molecular 'switch', regulating the access of incoming Nte peptide.

Another interesting protein diversity in Saf pili can be observed in Pro20 (located immediately after SafA$_{Nte}$). As revealed by SafDAA-*dsc* structure, Pro20 appears to play crucial roles in shaping the overall architecture and controlling the flexibility of Saf pili. In between SafD-SafA1, the proline adopts a *cis* configuration, and hence brings a minor structural variation in this region by introducing a hairpin-like Loop$_{G-A}$, which in turn enables the inter-molecular hydrogen bonding between subunits. In marked contrast, Loop$_{G-A}$ in between SafA1-SafA2 adopts an extended configuration, accompanied with the *trans* isomerization of Pro20. As a result, the SafA2 subunit is highly flexible and isolated, making no inter-molecular contact with the rest of the SafDAA-*dsc* structure. The movement of SafA2 gives rise to 'I'- or 'L'-like SafDAA architectures. It is reasonable to believe that the further variation of Pro20 isomerization and torsion angles will no doubt allow much versatile architecture of Saf pili that is necessary for the adherence and survival in different surfaces and environments. Consistently, as observed elsewhere, the isomerization of proline residue is frequently associated with protein folding (*Nicholson and De, 2011*; *Yuan et al., 1998*). Even within pilin protein families, the intrinsic flexibility of the proline can bring interesting structural diversities into subunit-subunit orientation/interaction, giving rise to different pili architecture and function (*Figure 8*). In *E. coli* Cfa pili, the isomerization switch of proline in Loop$_{G-A}$ allows CfaB-CafB to coil into a supra helical filament (*Li et al., 2009*) (*Figure 8C*). Consistently, similar proline isomerization can be observed in *E. coli* PapA-PapA and its supra helical oligomerization (*Hospenthal et al., 2016*) (*Figure 8E*). Even when the proline is in trans-isomerization, the minor difference/variation in the torsion angles of X-Pro-X (X for adjacent amino acid) can introduce significant movement/rotations of the subsequent pilin subunit (*Figure 8*). In summary, the SafDAA-*dsc* structure, together with previous observations (*Figure 8B–E*), might highlight a structural/functional role of proline residue in Loop$_{G-A}$, by the movement of which bacterial pili can assemble into diversified morphologies for adaptation of different enviroments (*Hung et al., 1996*; *Zav'yalov et al., 2010*).

### Host recognition by Saf pili

The SafDAA-*dsc* structure confirms the overall assembly of Saf pili, in which the major pilin SafA subunits, connected by the Nte peptides, is capped by a minor pilin SafD subunit. Previous studies showed Saf pili can mediate host regconition (*Carnell et al., 2007*). However, the exact host recognition mechanism remains elusive. In this report, the bindings were quantified by ELISA using antibody against recombinant Saf proteins with His-tag. Compared to the control SH3 protein, SafDAA-*dsc* displayed binding activity against host (*Figure 4AB*). As the recombinant pilus was restricted to SafDA-*dsc,* its binding activity/affinity was weaken. More dramatically, when the pilus was restricted to SafAA-*dsc*, SafD-*dsc* and SafA-*dsc*, the host recognition function was completely lost (*Figure 4B*), suggesting that (1) SafD and SafA are both required for host recognition; (2) SafD-SafA1 might be the most essential structural content/determinant for initial engagement; (3) the poly-SafAs might contribute to binding via a zip-in mechanism that enables intimate association between bacterium and host (*Figure 4C*). Until now, the host receptor of Saf pili is not yet defined. However, the host

receptors of *E.coli* AfaD/DraD adhesin/invasins, which shares ~30% identity with *Salmonella* SafD, have been reported. It has been shown that DraD and AfaD invasin subunits recognize the membrane bound integrins, possibly via the electrostatic protein:protein interaction (*Guignot et al., 2001*; *Zalewska-Piatek et al., 2008*). It is not clear whether SafD might utilize the similar receptor for recognition. Future biophysical and cellular experiments are required to characterize this in more detail.

## Biofilm formation by self-associating Saf pili

It has been reported that bacterial pili can facilitate micro-colony and biofilm formation (*Branda et al., 2005*; *Mandlik et al., 2008*). However, it is not clear how pili might mediate cell-cell interaction. In this report, for the first time, our data suggested that Saf pili can mediate biofilm formation (*Figure 7* and *Figure 7—figure supplement 1*). Unexpectedly, the SafDAA-*dsc* structures revealed remarkable Saf-Saf inter-cellular oligomerizations in the crystal (*Figure 5A–D*). Three Saf-DAA-*dsc* molecules form two sets of dimers (i.e. the Type I and Type II dimers) in a parallel head-to-tail configuration. The further intertwining of these dimers could give rise to a novel intercellular Saf-DAA-*dsc* trimerization. The SAXS scattering further verified the presence of multiple oligomerizations in solution (*Figure 5EF*). This is further supported by SEC-MALS characterization. The WT SafDAA-*dsc*, although mainly eluted as monomer, displayed a strong tendency of oligomerization/aggregation (*Figure 6B*). Consistently, the structure-based mutations, which target the dimeric interface and the proline reisdue (Pro20) in Loop$_{G-A}$, consistently disrupted the self-associating activity of SafDAA-*dsc* (*Figure 6B*). This has led to the design of the latex beads assay to demonstrate the self-associating activity in aggregation (*Figure 5—figure supplement 1*). In this system, the beads, mimicking bacteria in solution, were coated with WT SafDAA-*dsc*, mutants and BSA (the latter as control). Again, the beads assay was in good agreement with the crystallographic and biophysical observations.

More importantly, when Saf pili were overexpressed in *E. coli*, the cells displayed enhanced cellular aggregation and biofilm formation (*Figure 7* and *Figure 7—figure supplement 1*). In order to monitor the expression/surfacing of Saf pili, different techniques such as flow cytometry, EM and outer membrane extration were used (*Figure 7ABF*). All these results are consistent, suggesting a novel Saf-Saf inter-cellular oligomerization during cell-cell interaction and bacterial aggregation (*Figure 9*). Indeed, the self-associating inter-cellular oligomerizations are frequently observed (*Aricescu et al., 2007*; *Freigang et al., 2000*; *Himanen et al., 2010*). However, as estimated by SEC-MALS characterization, SafDAA-SafDAA association appears to be a weak interaction in the context of self-self dimerization/trimerization. This might be puzzling in the context of Saf pili as mighty cellular cross-linkers.

Recent studies showed that the inter-cellular polymerization/depolymerization of *H. infuenzae* self-associating Hap adhesin is an important mechanism to allow individual bacterium to traffic in/out of biofilm to survive in the host and to spread to other sites for colonization (*Meng et al., 2011*). Similar to Hap-Hap interaction, the weak interacting nature of SafDAA-SafDAA could be utilized by bacterium to adapt different environments. Furthermore, the lack of decisive disruption by structure-based mutations also implies that Saf pilus, which contains hundred of SafA subunits, could adopt multiple Saf-Saf inter-cellular engagements, which includ (but not restricted to) the oligomerization unveiled by the SafDAA-*dsc* structures. Supportively, similar self-associating activity/oligomerization have been reported in *E. coli* ECP pili, *E. coli* Ag43 adhesin and *S.parasanguis* Fap1 adhesin (*Garnett et al., 2012a*; *Garnett et al., 2012b*; *Heras et al., 2014*).

## Conclusion

SafD, which is highly conserved in *Salmonella* (*Figure 1—figure supplement 1*), displayed vaccination potential in mice model. In this study, we have determined the crystal structure of SafD-*dsc*. The recombinant production and structural information might lay the important foundation for future structure-based vaccination design (*Kulp and Schief, 2013*). In addition, our biochemical and structural characterizations of SafDAA-*dsc* might help to define important functions of Saf pili. Firstly, cell adherence assay with SafDAA-*dsc* and derivatives demonstrated that Saf pili are poly-adhesive and could contribute to host recognition via a zip-in mechanism. Secondly and more importantly, biophysical characterizations together with the bead/cell aggregation and biofilm formation assays

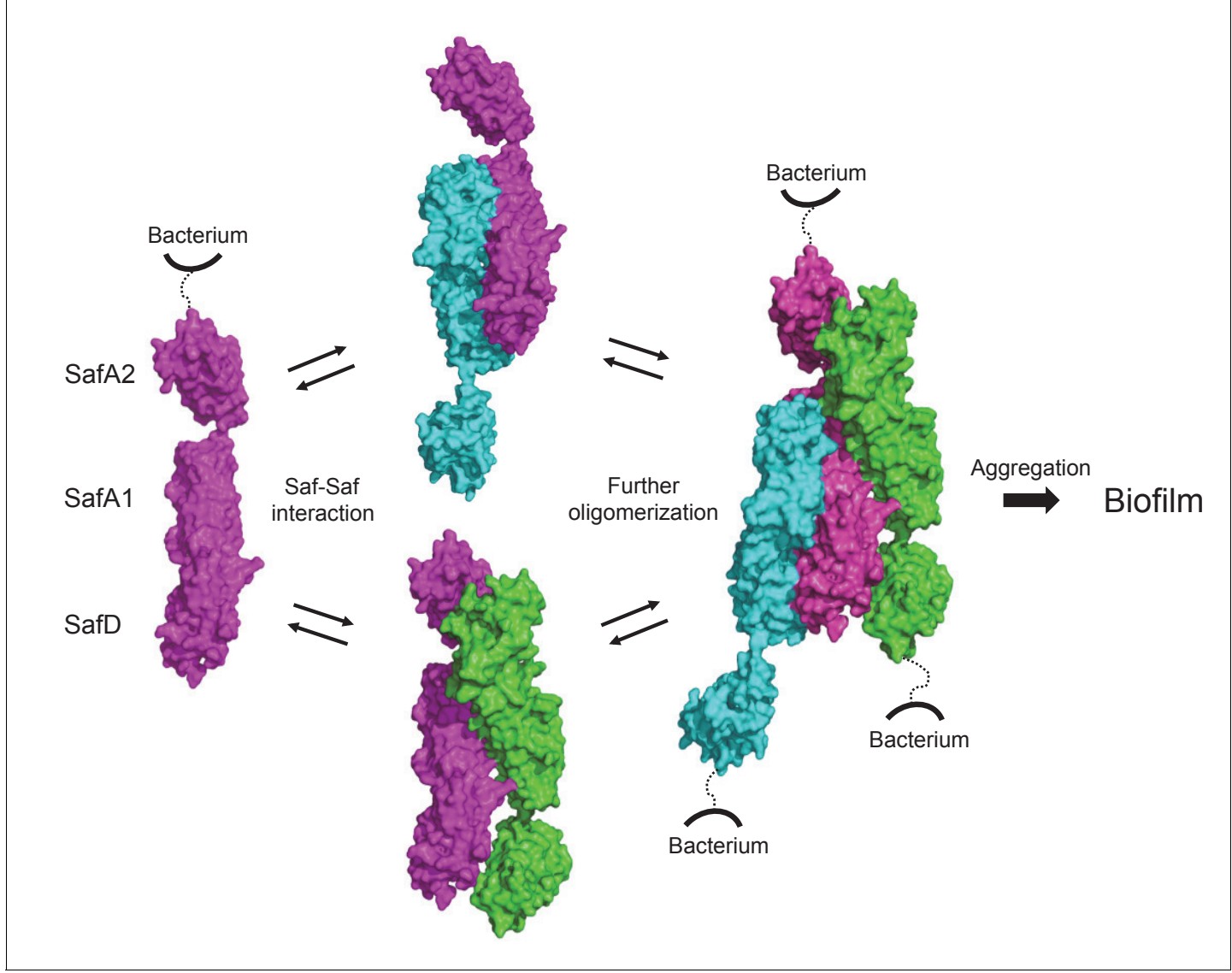

**Figure 9.** Structural and functional characterization of SafDAA-*dsc* reveal a novel Saf-driven self-associating mechanism for bacterial aggregation and biofilm formation. In the initial stage, Saf pili can mediate cell-cell interaction via versatile SafDAA dimerizations. As the inter-cellular oligomerization proceeds further, Saf pili might intertwine against each other, enabling a higher order Saf pili oligomerization (n > 3) to generate necessary attractive forces for bacterial aggregation and ultimate biofilm formation.

DOI: https://doi.org/10.7554/eLife.28619.023

uncover a novel biofilm formation function of Saf pili. As supported by the SafDAA-*dsc* structure and subsequent mutation characterization, SafDAA-SafDAA dimerizations are critical in self-associating activity and biofilm development. This, together with previous observations in Hap-Hap, Ag43-Ag43, ECP-ECP and Fap1-Fap1 etc, reiterates the idea that the self-associating inter-cellular oligomerization might represent a simple, but powerful strategy for bacterial aggregation and biofilm formation.

## Materials and methods

### Molecular cloning

The DNA fragments encoding SafDA-*dsc* and SafDAA-*dsc* were synthesised and inserted into pET-15b vector (Novagen) using restriction sites *Nde*I and *Xho*I. The cDNA of SafD-*dsc* was obtained by

PCR using pET-15b-SafDA-*dsc* as template. The molecular cloning steps such as DNA gel extraction/purification, restriction digestion and ligation were performed with standard protocols.

To obtain *SafABCD* gene cluster, we extracted the genomic DNA from *Salmonella enterica* serovar Typhimurium 14028 strain (a kind gift from Prof. Yu-feng Yao, Shanghai Jiao Tong University). The plasmid that enables expression, surfacing and assembly of Saf pili was obtained using a seamless cloning strategy. Briefly, a pair of primers were designed to amplify the complete *SafABCD* gene cluster (encoding the mature SafA without signal peptide, and the intact SafB, SafC and SafD). The resulted PCR product contained 20 base pairs of homology sequence with pASK-IBA4 vector (IBA) in its 5'- and 3'-end. The vector was then digested with *Bsa*I (New England Biolabs). The PCR product was ligated into a *Bsa*I-digested pASK-IBA4 vector according to the manufacturer's protocol described in the Quick-Fusion cloning kit (Selleck, Houston, Texas). The mutants of pASK-IBA4-Saf were engineered using QuickChange mutagenesis kit (Agilent technologies).

## Protein expression, purification and crystallization

The pET-15b-SafD-*dsc* and pET-15b-SafDAA-*dsc* plasmids were transformed into *E.coli* BL-21 (DE3) strain, respectively. Cells were grown at 37°C in LB culture medium, supplemented with 100 µg/ml ampicillin. When $OD_{600}$ reached ~0.6, bacteria were induced with 1 mM IPTG followed by further incubation overnight at 22°C. The culture was harvested by centrifugation (4000 rpm, 20 min). The cells were resuspended in 20 mM Tris pH 8.0 and 100 mM NaCl buffer and lysed using a high-pressure homogenizer. The cell lysate was centrifuged at 20,000 rpm for 1 hr. The clear lysate was collected and filtered through a 0.22-µm filter (Millipore). The clear lysate containing SafD-*dsc* was loaded onto a HisTrap column (GE healthcare). The eluate was pooled and incubated with thrombin protease (Invotrogen). To remove the cleaved His tag, the protein sample was reloaded onto the HisTrap column. The flow-through was collected and dialyzed against a buffer of 20 mM Tris pH 8.0 and 20 mM NaCl overight at 4°C before it was further purified by anion-exchange chromatography using a Q column (GE healthcare). Finally, to polish the final purity, a gel filtration step using S100 sepharose column (GE healthcare) was performed. As for SafDAA-*dsc*, a similar purification protocol including the applicaitons of Ni-chelating affinity chromatography (His-Trap), anion exchange chromatography (Q), hydrophobic interaction chromatography (Phenyl) and gel filtration (S100) was used. The final purities of SafD-*dsc* and SafDAA-*dsc* were analyzed using SDS-PAGE.

For crystallization, SafD-*dsc* and SafDAA-*dsc* were concentrated to 50 mg/ml and 45 mg/ml, respectively. JBScreen classic 1–10 kits (Jena Bioscience) together with hanging-drop vapor-diffusion method were used for initial crystallisation screen at 20°C. The hit condition of SafD-*dsc* crystals contains a reservoir solution of 25% PEG 8000, 20 mM LiCl, pH 5.0. The crystallisation condition of SafDAA-*dsc* is 10% PEG 4000, 10% 2-Propanol, 100 mM Sodium Citrate, pH 5.6.

## Data collection, phasing and structure refinement

Crystals were flash cooled to 100 K in liquid nitrogen using 20% PEG 400 as cryoprotectant. X-ray diffraction data were collected on beamline BL17U of Shanghai Synchrotron Radiation Facility (Shanghai, China). Crystals of SafD-*dsc* diffracted to 2.2 Å and were in space group $P2_12_12_1$ with cell dimensions of a = 32.5 Å, b = 49.7 Å and c = 148.8 Å. The SafDAA-*dsc* crystals diffracted to 2.8 Å and were in space group *C*2. The unit cell parameters were a = 133.3 Å, b = 66.1 Å, c = 187.7 Å and β = 96.2°. The diffraction data were processed, integrated and scaled using MOSFLM/SCALA (*Collaborative Computational Project, Number 4, 1994*) (RRID:SCR_007255). The statistics of the data collection are shown in *Table 1*.

For SafD-*dsc* phasing, molecular replacement using AfaD (PDB code: 2IXQ) as search model was used. The coordinates of AfaD were pruned by sequence alignment and CHAINSAW (*Collaborative Computational Project, Number 4, 1994*) before they were supplied to program PHASER (*Collaborative Computational Project, Number 4, 1994*). When searching the first SafD molecule, the best solution showed a Z-score of 4.0 after translation function. However, when the second SafD was placed correctly in the crystal, a high Z-score value (>10) was etimated, suggesting a clear solution. Intermittent manual building implemented in COOT (*Collaborative Computational Project, Number 4, 1994*) was combined with structure refinement using REFMAC5 (*Collaborative Computational Project, Number 4, 1994*) to improve the initial models produced by PHASER.

As for SafDAA-*dsc*, the refined SafD-*dsc* and the published SafA (PDB code: 2CO4) structures were used as search templates. The intial phases were estimated by molecular replacement using PHASER for further model building and refinement as described above.

The structures of SafD-*dsc* and SafDAA-*dsc* was refined by conjugate gradient minimization implemented in REFMAC5 (*Collaborative Computational Project, Number 4, 1994*) and simulated annealing implemented in PHENIX.REFINE (*Adams et al., 2010*) (RRID:SCR_014224). The B-factors were refined with TLS corrections (*Winn et al., 2001*). The final model of SafD-*dsc* contains 277 protein residues and 125 water molecules. The final model of SafDAA-*dsc* contains 1275 protein residues and 50 water molecules. Ramachandran statistics calculated by PROCHECK (*Laskowski et al., 1993*) indicate that 98.5% atoms of SafD-*dsc* are in the most favored region, and 1.1% are in the allowed regions. As for SafDAA-*dsc*, 95.6% atoms are in the most favored region, and 4.1% are in the allowed regions. The detailed structure refinement statistics are reported in *Table 1*. Coordinate of SafD-*dsc* and SafDAA-*dsc* have been deposited into the Protein Database Bank. The entry codes are 5Y9G and 5Y9H.

## Latex beads assay

The latex beads assay was carried out using published protocol (*Meng et al., 2011*) with a minor modification. After rinsed with coupling buffer (50 mM MES, pH 5.2), 2 µm polystyrene latex beads (Sigma) were incubated with 5 µg protein at room temperature for 24 hr. The coupling buffer was then used to wash the beads three times. The beads coated with SafDAA-*dsc*, SafD-*dsc*, SafA-*dsc* or BSA proteins were resuspended in 100 µl PBS. After a 6 hr incubation, the SafDAA-driven aggregation was visualized by phase-contrast microscopy (Nikon Eclipse Ti).

## Flow cytometry analysis

DH5α strains harboring plasmid pASK-IBA4 or pASK-IBA4-Saf, were induced by 2% anhydrotetracycline (AHT) overnight at 37°C in LB medium. Bacteria were havested by centrifugation (4000 rpm, 5 min) followed by wash using PBS buffer. The same washing protocol was also included in diffenent experimental steps as described below. The cells were fixed with 500 µl 4% paraformaldehyde for 30 min before they are resuspended in 500 µl PBS, 0.1% BSA and 0.5 µg/ml rabbit anti-Strep polyclonal antibody (GenScript, RRID:AB_915541) for 1 hr. Cells were then incubated with 200 µl PBS, 0.1% BSA and FITC-conjugated secondary antibody (1:100, BBI life sciences) for 1 hr before they were subjected to flow cytometry analysis (BD LSRFortessa). 2 µm fluorescent blue polystyrene latex beads (Sigma) were used as size reference to ensure the selection of single bacteria for subsequent Saf surfacing analysis. The flow cytometry data were processed with FlowJo software (RRID:SCR_008520).

Furthermore, it has been shown that flow cytometry could be used to monitor bacterial aggregation (*Mackenzie et al., 2010*; *Torrent et al., 2012*). The cells with or without Saf pili were grown in DMEM medium before they were subjected to flow cytometry analysis using the similar protocol as described above. In these experiment, 2 µm fluorescent blue polystyrene latex beads (Sigma) were also used as size reference. The cell aggregations were registered with high values of FSC and SSC signals. The percentages of bacterial aggregation in Saf- or control-cells were quantified using FlowJo software (RRID:SCR_008520).

## Immunoelectron microscope analysis

*E. coli* DH5α cells harboring Saf pili expression plasmids were induced by 2% AHT at 37°C in LB medium. Bacteria cultures (200 µl) were fixed with equal volume of 4% paraformaldehyde for 30 min. The cells were collected by centrifugation (3000 rpm, 5 min), followed by resuspension and wash in PBS buffer. After blocking of PBS buffer containing 5% BSA, the cells were incubated with primary rabbit anti-Strep polyclonal antibody (1:100 diluted in PBS, 1% BSA, GenScript, RRI D: AB _9155 41) for 30 min at room temperature. The cells were washed six times by PBS buffer, 1% BSA, and then reacted with secondary 20 nm gold-labled particle coated with anti-rabbit IgG antibody (1:100 diluted in PBS buffer, 1% BSA) for 30 min at room temperature. The cells were then subjected to vigourous wash before they were transferred onto the copper grid (Sigma). Images were recorded by JEOL JEM1230 electron microscope.

## Western blot analysis of bacterial outer membrane

*E.coli* DH5α cells containing pASK-IBA4-Saf and mutants were induced by 2% AHT at 37°C in LB medium. Bacterial outer membrane proteins were obtained on the basis of sarkosyl insolubility (*Carlone et al., 1986*). Bacteria were havested by centrifugation followed by wash using 20 mM Tris pH 8.0 buffer. The cells were broken by sonication. To remove the unbroken cells, centrifugation (3000 g, 15 min) was used and the supernatant was subjected to further centrifugation (100,000 g, 1 hr) to harvest total membranes. The membranes were resuspended in 20 mM Tris pH 8.0 buffer containing 0.5% N-lauroylsarcosine. After rocking at room temperature for 5 min, the sample was centrifugated at 100,000 g for 1 hr. The outer membrane was resuspended in 20 mM Tris pH 8.0, 150 mM Nacl buffer containing 1% Zwittergent 3–12, followed by rocking at room temperature for 30 min. The outer membrane protein was finally isolated by centrifugation at 100,000 g for 1 hr. The outer membrane proteins were resolved by SDS-PAGE and then electrotransferred to polyvinylidene difluoride (PVDF) membrane (Millipore). The membrane was blocked by 5% milk for 1 hr at room temperature, followed by overnight incubation with PBS buffer containing 3% BSA and rabbit anti-Strep polyclonal antibody (1:4000, GenScript, RRID:AB_915541) at 4°C. After washing with PBST buffer, the membrane was incubated with PBS buffer containing 3% BSA and HRP-conjugated donkey anti-rabbit IgG secondary antibody (1:20000, BBI life sciences) for 1 hr at room temperature. Specific bands were detected with chemiluminescense HRP substrate (Millipore) and visualized by Amersham imaging system (GE healthcare).

## Size exclusion chromatography - multi-angle light scattering

SafDAA-*dsc* and mutants were subjected to gel filtration analysis using a WTC-015S5 sized exclusion column (Wyatt Technology). The elution of each sample was analyzed by a 1260 infinity liquid chromatography system (Agilent Technology) linked with inline DAWN HELEOS-II MALS and Optilab rEX differential refractive index detectors (Wyatt Technology). For each run, a 40-μl sample (2.5 mg/ml) was injected. The sample was eluted at a flow-rate of 0.5 ml/min for SafDAA-*dsc* and mutants. Data were recorded and analyzed using ASTRA VI software (Wyatt Technology).

## Small angle X-ray scattering

SafDAA-*dsc* and mutants in the buffer containing 20 mM Tris, 100 mM NaCl, pH 8.0 were concentrated to 1, 2.5 and 5 mg/ml before SAXS characterization. SAXS data were recorded at Beamline station BL19U2 (Shanghai Synchrotron Radiation Facility, SSRF, China) under a X-ray beam (wavelength, 1.03 Å) and recorded for a total $q$ range from 0.01 to 0.35 Å$^{-1}$. Data subtraction and analysis were performed using PRIMUS (*Petoukhov et al., 2012*). Crystal data fitting was done with the OLIGOMER algorism implemented in CRYSOL (*Petoukhov et al., 2012*). As stated in *Table 1*, there are three SafDAA-*dsc* molecules in ASU. In the analysis of crystal fitting, six sets of SafDAA-*dsc* coordinates including three different monomers, two possible dimers and one trimer were used to fit the experimental SAXS data, leading to the determination of each fraction of the coordinates in the solutions of 1, 2.5 and 5 mg/ml. For the purpose of conciseness, only the result of 1 mg/ml SafDAA-*dsc* is reported in *Figure 5EF*.

## Cell aggregation assay

The plasmids of pASK-IBA4-Saf and mutants were transformed into *E. coli* DH5α strains. The cells were growned at 37°C in LB medium with shaking until OD600 = 0.6. 10 μl bacteria culture was mixed with 90 μl DMEM medium (Gibco) followed by static induction (i.e. no shaking) by 2% AHT in 96-well plate for 24 hr. The cell aggregation was photoed by phase-contrast microscopy (Nikon Eclipse Ti).

## Biofilm assay

Biofilm formation was quantified with crystal violet staining (*Berry et al., 2014*; *Ulett et al., 2007*). The pASK-IBA4-Saf construct encoding the assembly and surfacing of the complete Saf pili was transformed into DH5α strains. The cells were growned at 37°C until OD$_{600}$ = 0.6. 10 μl bacteria culture was inoculated to 90 μl DMEM medium (Gibco) containing 2% AHT in a 96-well plate. The cells in DMEM were further incubated at 37°C (without rocking) for 36 hr. The 96-well plate was then gently washed with PBS buffer and fixed with methanol for 15 min. Crystal violet staining solution

(BBI life sciences) was used for bacterial staining. 100 µl 33% acetic acid was added to dissolve crystal violet stain derived from bacterial bioflm. The biofilm formation acitivties of Saf pili and mutant were quantified by $OD_{570}$.

## Cell adherence assay

The adhesive activities of recombinant SafDAA-*dsc* and derivatives were quantified by the cellular ELISA assay described in *Laarmann et al. (2002)*. IPEC-J2 (RRID:CVCL_2246) cells were obtained from the Deutsche Sammlung von Mikroorganismen und Zellkulturen (DSMZ). The identity had been authenticated by short tandem repeat DNA profiling. Cells were regularly tested to ensure none mycoplasma contamination. IPEC-J2 cells were seeded in a 96-well plate at a density of $2.5 \times 10^4$ cells per well. When the cell culture reached ~100% confluence after overnight incubation, the IPEC-J2 cells were washed twice with PBS buffer and fixed by 4% paraformaldehyde for 15 min. Subsequently, the treated cells were blocked with 3% BSA to prevent non-specific interaction, followed by further incubation with SafDAA-*dsc* and deriviatives at room temperature. After wash, the cells were incubated with PBST buffer containing 1% BSA and the anti-His antibody (Thermo Fisher Scientific, RRID:AB_2533309) for 1 hr. The cells were then washed by PBST buffer and incubated with a HRP-conjugated secondary antibody (Jackson ImmunoResearch) for 1 hr. After thoroughly wash, the TMB substrate (Cell signaling technology) was used to generate detectable signal. Stop reagent (Cell signaling technology) was added to stop the reaction. The absorbance at 450 nm were used to monitor the adherence activities of recombinant Saf pili (i.e. SafDAA-*dsc*) and derivatives.

## Acknowledgement

This work was supported by research grants 81770142, 81370620, 81570120, 31070645 from National Scientific Foundation of China (to GM), 'The Program for Professor of Special Appointment (Eastern Scholar) at Shanghai Institute of Higher Learning' (to GM), a research grant 20152504 from 'Shanghai Municipal Education Commission—Gaofeng Clinical Medicine Grant Support' (to GM), a research grant 11JC1407200 from SMSTC (to GM), a research grant 12ZZ109 from SME (to GM), We thank the staffs from BL17U/BL18U1/BL19U1/BL19U2/BL01B beamline of National Center for Protein Sciences Shanghai (NCPSS) at Shanghai Synchrotron Radiation Facility (SSRF), for assistance during data collection. We thank reviewers, Drs. Scott Hultgren, Han Remaut and Steve Matthews, for their insightful and helpful suggestions. We also thank Dr. Haiyan Wu for her kind assistance in flow cytometry analysis.

## Additional information

### Funding

| Funder | Grant reference number | Author |
| --- | --- | --- |
| National Natural Science Foundation of China | 81370620 | Guoyu Meng |
| National Natural Science Foundation of China | 81570120 | Guoyu Meng |
| National Natural Science Foundation of China | 31070645 | Guoyu Meng |
| Shanghai Municipal Education Commission | The Program for Professor of Special Appointment (Eastern Scholar) | Guoyu Meng |
| Shanghai Municipal Education Commission | Gaofeng Clinical Medicine Grant Support (20152504) | Guoyu Meng |
| Shanghai Municipal Education Commission | 12ZZ109 | Guoyu Meng |
| Shanghai Science and Technology Commission | 11JC1407200 | Guoyu Meng |
| National Natural Science Foundation of China | 81770142 | Guoyu Meng |

The funders had no role in study design, data collection and interpretation, or the decision to submit the work for publication.

## Author contributions

Longhui Zeng, Li Zhang, Pengran Wang, Data curation, Formal analysis, Investigation, Methodology; Guoyu Meng, Conceptualization, Resources, Data curation, Formal analysis, Supervision, Funding acquisition, Validation, Investigation, Visualization, Methodology, Writing—original draft, Project administration, Writing—review and editing

## Author ORCIDs

Guoyu Meng (iD) http://orcid.org/0000-0001-7904-2382

## Decision letter and Author response

Decision letter https://doi.org/10.7554/eLife.28619.032
Author response https://doi.org/10.7554/eLife.28619.033

## Additional files

### Supplementary files

• Transparent reporting form
DOI: https://doi.org/10.7554/eLife.28619.024

### Major datasets

The following datasets were generated:

| Author(s) | Year | Dataset title | Dataset URL | Database, license, and accessibility information |
|---|---|---|---|---|
| Zeng LH, Zhang L, Wang PR, Meng G | 2017 | Crystal structure of Salmonella SafD adhesin | http://www.rcsb.org/pdb/explore/explore.do?structureId=5Y9G | Publicly available at Protein Data Bank (accession no: 5Y9G) |
| Zeng LH, Zhang L, Wang PR, Meng G | 2017 | Crystal structure of SafDAA-dsc complex | http://www.rcsb.org/pdb/explore/explore.do?structureId=5Y9H | Publicly available at Protein Data Bank (accession no: 5Y9H) |

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
