## [Decision Letter]

Thank you for submitting your article "Structural basis of host recognition and biofilm formation by *Salmonella* Saf pili" for consideration by *eLife*. Your article has been favorably evaluated by Gisela Storz (Senior Editor) and three reviewers, one of whom, Scott Hultgren (Reviewer #1), is a member of our Board of Reviewing Editors. The following individuals involved in review of your submission have agreed to reveal their identity: Han Remaut (Reviewer #2); Steve Matthews (Reviewer #3).

The reviewers have discussed the reviews with one another and the Reviewing Editor has drafted this decision to help you prepare a revised submission.

Summary:

Zheng and co-workers report the X-ray structures of the *Salmonella enterica* Saf pilus subunits and describe their role in cell adhesion and self-aggregation. The two reported structures are that of the SafD tip subunit complemented with the SafA donor strand, and a concatenated construct comprising SafD-SafA-Saf-SafNte. The structural information is novel and of high quality. Two alternative conformations are seen for the SafD-A-A polymer and this minimal pilus structure is seen to align longitudinally in the crystals, which the authors suggest to hint at a possible role in bacterial self-interaction and biofilm formation. This hypothesis is evaluated by an in vitro bead agglutination assay and a crystal violet biofilm formation assay in *E. coli* recombinantly expressing the saf operon.

Essential revisions:

1) The relative contribution / requirement of SafD needs to be established in the cell-based adhesion assay. This can be done by repeating the assay with a SafAA construct.

2) The authors need to demonstrate recombinant Saf pili are expressed in their *E. coli*-based assays. The flow cytometry is flawed as done now because of the presence of the 50 mM EDTA. Saf pilus formation also needs confirmation by electron microscopy.

3) The bead assay needs to be made quantitative. How representative are the current images? Why use a bead assay if the authors can look at bacterial clustering directly using light microscopy?

I agree with reviewer 3 that a better alternative to the bead assay is solution-based affinity measurements for SafAAD self-association. By BLI, ITC or SPR and possibly supplemented with SAXS.

4) If using the biofilm assay to compare point mutants, the authors need to evaluate whether the mutants do not alter Saf expression on the cell surface.

5) The authors need to address whether these observations are artifacts of crystallography. By definition, crystals are built using intermolecular contacts, including from hydrogen bonds and electrostatic interactions. Furthermore, conformational changes can be induced in order to propagate an ordered crystal. Are the contacts and observations artifacts or real?

6) The authors describe biological biofilm-formation assays that fits with their structural work and proposed function roles, but the manuscript lacks solution-based, biophysical characterisation that might support the effects of the proline peptide bond and the Saf-Saf contacts i.e. NMR, SAXS, native MS for example.

Below are the verbatim comments of the reviewers from which the above were distilled. We reproduce these here for your benefit although the most critical comments for your attention are those listed about under "essential".

*Reviewer #1:*

In this manuscript, Meng and colleagues present new structural and functional data outlining the structural basis for Saf pilus self-association. They first present an X-ray crystal structure of a self-complemented SafD single-domain adhesin which, like other members of the Afa/Dr poly-adhesin subfamily to which SafD belongs, reveals a classic Ig-like fold and a small negatively-charged patch that they hypothesize may correspond to a receptor binding site. Most notably, they observe that the Leu4 residue in the N-terminal donor strand may rest within the P** pocket of SafD, interacting with the hydrophobic core in a similar manner to the previously-characterized Phe3 residue.

The authors next present a structure of the SafDAA tip-like structure, which adopts two distinct conformations within the crystallographic asymmetric unit defined by a 120-degree shift in the orientation of the proximal SafA subunit relative to the penultimate SafA subunit. The authors posit that the SafDAA isomerization state of Pro20 within the A-A linker regulates access to these two conformational states. The authors also demonstrate that the SafD-A interaction is likely fairly rigid due to a series of hydrophobic interactions between the two subunits.

The authors demonstrate that both SafD and SafA are both required for cell adherence, and that the binding affinity of SafDAA is greater than that of SafD alone. The authors thus conclude that the Saf pilus utilizes a polyadhesive mechanism to interact with host receptors.

Finally, the authors probe the role of the two observed SafDAA structures in pilus self-association and biofilm formation. In the so-called Type I dimerization, the authors demonstrate that a series of hydrogen bonds between SafA and SafD are required to mediate dimerization, while in Type II dimerization, close approximation of complementary surfaces between all three Saf subunits lead to an interaction. This Type II dimerization can be abrogated by restricting the torsion angles of residue 20 to prevent the large conformation changes required for this approximation to take place.

Originality and Significance

While the authors do not attempt to identify the molecular receptor or tissue tropism of the Saf pilus, their structural and functional insights into the mechanism of self-association are both interesting and novel. The significance of this work could be augmented by further discussion of the generalizability of these findings to other pilus systems by comparing the sequences and structures of SafD and A to sequences and structures of similar pilus types. Additionally, the contribution of the Leu4 residue to the DSE interaction could be interesting if found in other pilus systems.

1) There are many grammatical errors throughout the manuscript that should be corrected before publication.

2) The authors suggest that the SafD binding site is located at an electronegative patch on the side of the SafD molecule, but fail to discuss this further. These residues could be highlighted on Figure 2E to allow for comparison to the known binding residues of other members of the AfaD family. There was no attempt to mutate any of the residues making up this surface to probe their role in binding to IPEC-J2 cells assessed.

3) The increase in binding affinity of the SafDAA construct vs, the SafDA construct could either mean that the second SafA subunit interacts with the host receptor directly, or that its presence results in subtle conformation changes that alter the affinity of the two terminal Saf subunits for its ligand. The superposition of the two SafD structures is found in Figure 2D; however, this analysis is limited to the conformation of Tyr13. Are there other structural changes in SafD? Similarly, are there any SafA side chains whose positions differ significantly in the I and L forms, either within each conformation or between the two conformations that may indicate structural changes underlying the affinity difference?

4) The I/ for the outer resolution shell of the SafDAA-*dsc* structure is only 0.8. Without seeing the density or knowing the CC1/2 in this shell, a resolution cutoff of 2.6 Å may not be appropriate. The B-factors and high R_free_ in the outer shell are also concerning.

Reviewer #2:

Zheng and co-workers report the X-ray structures of the *Salmonella enterica* Saf pilus subunits and describe their role in cell adhesion and self-aggregation. The two reported structures are that of the SafD tip subunit complemented with the SafA donor strand, and a concatenated construct comprising SafD-SafA-Saf-SafNte. The structural information is novel and of high quality. Two alternative conformations are seen for the SafD-A-A polymer and this minimal pilus structure is seen to align longitudinally in the crystals, which the authors suggest to hint at a possible role in bacterial self-interaction and biofilm formation. This hypothesis is evaluated by an in vitro bead agglutination assay and a crystal violet biofilm formation assay in *E. coli* recombinantly expressing the saf operon.

The study is of potential broad interest, but needs attention in a number of points. The weak point of the study is that the role of Saf pili is not directly studied in *Salmonella enterica* itself and that the current data on the bead agglutination and *E. coli* biofilm formation are inconclusive.

Hence, a more modest discussion and representation is needed where extrapolating in vitro and structure-based findings to the in vivo context of *Salmonella enterica* infections. This begins with a more precise representation of the available literature on previous studies on the role of saf and other fimbriae in *Salmonella enterica* pathogenesis and biofilm formation. To confirm the claims on the Saf pilus adherence and self-aggregation, additional data would be needed as detailed in the point-by-point review.

Abstract:

"Although Saf pilus is ubiquitously expressed in *Salmonella enterica*, its structure and function remain poorly understood" – this is a strong claim without reference, are Saf pili ubiquitously expressed? Presence in the genome does not equal expression. In general, the available literature on Saf biology and previous functional studies on fimbriae in *Salmonella* pathogenesis need to be better described in the introduction. The claim is repeated in the Introduction, again without any reference.

"More importantly, Saf-Saf oligomerization helps to define a universal, inter-cellular self-associating mechanism in bacterial aggregation/colonization." What is the universal self-association mechanism that is proposed?

Introduction:

"Although Saf pili are ubiquitously expressed in *Salmonella enterica*, it is not clear how Saf pili mediate host recognition, and whether they participate in cell-cell interaction and biofilm development. In this study, we aim to provide more structural and functional insights into this important pilus subtype." Any claims on the role or importance of Saf pili in *Salmonella* need references or new experimental data.

Results:

"Furthermore, based on sequence alignment (Figure 2 and Figure 1—figure supplement 1), SafD is thought to be adhesive." This is a non-logical claim as currently presented. If referring to sequence homology with *E. coli* AfaD, is it expected that SafD acts as an invasin? This should then be specified and included in the Discussion.

"However, in P* position (a favored interaction that catalyze/stabilizes the exchanged product during DSE), Fhe3 is no longer in direct interaction with SafD." "Fhe3" should be corrected to "Phe3" throughout manuscript. What is the "catalytic" role the authors are referring to?

"The superimposition of the SafD molecules shows interesting structural variations/flexibilities in LoopA-B on top of the P5 pocket, implying this loop might play a regulatory role in DSE (Figure 2)." Why does the presence of two conformations in the LoopA-B imply a regulatory role? What kind of regulatory role do the authors have in mind and evidence is there to invoke such role?

"The SafD, SafA1, SafA2 and SafA3Nte have assembled into a thread-like shape with 26 Å in diameter and 136 Å in length/height (Figure 3 and Figure 3—figure supplement 1)." Why are linker sequences not shown and are N- and C-termini labelled in the Saf subunits? In the supplementary figure the linkers are shown.

"In between SafD and SafA1, Pro20 adopts a cis-configuration, enabling a kink in the subunit-subunit linker loop that, in turn, allows the formation of intra-molecular hydrogen bonds (Figure 3—figure supplement 1)." Do the authors mean "intermolecular"? Also, what is shown in Figure 3—figure supplement 1? The labelled residues do not make sense. T111 looks like an Ile, or did the authors include the -OH hydrogen? Q134 and N6 do not make sense as Q or N, or again were the -NH2 hydrogens included? If including hydrogens, that should be done for all residues and all atoms in a residue; or clearly specified in the legend. If they are hydrogens, the Q134 side chain amide needs a 180 degree rotation.

"Collectively, these data have suggested a poly-adhesive activity of Saf pili, in which SafD and SafA subunits can bind concertedly to the unknown host cell receptors to enable intimate host:bacterium interaction and colonization via a zip-in mechanism (Figure 4)." What is the contribution of SafD in the polyadhesive binding? Does a SafA polymer or SafAA bind the IPEC-J2 cells?

"As shown in Figure 5, the beads coated with the recombinant SafDAA-*dsc* protein aggregated together to form clusters." The authors use a bead agglutination assay to evaluate the self-association of SafDAA-*dsc*. The experiments as shown are not conclusive. How representative are the images? A more quantitative assay or representation of the bead assay is required to draw conclusions.

For the *E. coli* cell sorting and biofilm assay (Figure 6), it is not currently demonstrated that surface-exposed Saf pili are indeed expressed (see below comments in the Materials and methods) and for the evaluation of the mutants (Figure 6), how do the authors differentiate whether reduced biofilm titers stem from reduced self-association versus reduced Saf pilus expression? If surface-exposed Saf pili are formed, are the amounts obtained from recombinant plasmid-based expression representative for those seen under native conditions in *Salmonella*? Is the N-terminal Strep tag on SafA in the recombinant saf operon compatible with Saf pilus formation?

"At the initial stage, Saf pili might interact with each via Type I dimerization. As biofilm develops, the Saf pili can proceed into Type II dimerization, by each step of which the two adjacent cells move ~40 Å towards each other, giving rise to further intimate association and aggregation, leading to the ultimate micro-colony and biofilm formation (Figure 8)." It is not clear what the sequential model is based on. Why would type I interactions take priority on Type II interactions?

Discussion:

"The movement of LoopA-B and Try13 sidechain appears to act as a molecular "switch", allowing or precluding the access of incoming Nte peptide" There is no experimental basis for this claim – see also comment above.

"It is reasonable to believe that the further variation of Pro20 isomerization and torsion angles will no doubt allow much versatile architecture of Saf pili that is necessary for the adherence and survival in different surfaces and environments." It is not clear what is meant here. Pro will be *cis* or *trans*, no further variation, and the dihedral angles of X-Pro-X bounds have reduced conformational freedom compared to that of other residues. Also what is meant with the concluding remark on the previously unrecognized role of proline in the G-A loop is unclear.

"The cell adherence assay presented in this report showed that both SafA and SafD subunits are important for the binding of host cell receptors." The current data do not proof the involvement of SafD in the adherence – see comment above. Also the claim "The mini Saf pilus, SafDAA-*dsc*, appeared to mimic the role of Saf pili" should be reworded. There is no data on the Saf pilus presented, or referred to in the Introduction.

"Based on sequence alignment, the host receptors integrin beta3 and fibronectin of AfaD (~30% sequence indentity with SafD) might be possible targets for Saf pili that should be further investigated in future studies (Cota et al., 2006)." The reference shows AfaD binding to integrin beta1. Also, there is several studies showing that pilus adhesins with seq. ID's well above 50% percent can have unrelated receptors. Is there evidence that SafD acts as invasion similar to *E. coli* AfaD?

Materials and methods:

Flow cytometry analysis is used to evaluate the surface expression of Saf pili on *E. coli*. The Materials and methods describe use of 50mM EDTA for fixing and antibody based staining of the cells. EDTA treatment permeabilizes the outer membrane. How can the authors ensure that surface-localized Saf pili were formed? Also, if Saf pili induce self-aggregation, is the flow-cytometry experiment looking at single cells or cell clusters? This should be apparent from the scattering levels.

*Reviewer #3:*

The paper describes the high resolution structural insight into a biopolymer of the Saf pilus from *S. enterica*. The Saf pilus is a member of the chaperone-usher pathway which is probably one the best characterised pilus assembly systems. The authors use the self-complementing approach, which has been previously been adopted to create single molecule mimics of the polymer. The crystal structures are described for the SafD tip adhesin and a three-subunit polymer of SafD-SafA-SafA. A conserved proline is seen in both *cis* and *trans* conformations depending on whether it is between two SafA subunits or SafA and the SafD tip. Interestingly, the *cis* proline alters the overall architecture of the pilus and introduced a pronounced bend. They also reveal an interesting network of intermolecular contacts within the crystal that suggest a new mode of bacteria-bacteria interactions. These novel features are probed with mutagenesis and biofilms assays.

The paper is reasonably well written, and the figures are clear and professionally prepared. With a further proof read and review of the figures, the quality of the presentation would fit with the high calibre of this journal and other papers in this field. The structures of the pilus subunits are known and therefore this aspect does not add new insight. However, the molecular detail of the proline peptide bond and the Saf-Saf contacts are potential novel and of interest to the chaperone-usher field and those interested in bacterial-host recognition and biofilms.

One major question the authors need to address is whether these observations are artefacts of crystallography. By definition, crystals are built using intermolecular contacts, including from hydrogen bonds and electrostatic interactions. Furthermore, conformational changes can be induced in order to propagate an ordered crystal. Are their contacts and observations artefacts or real?

The authors describe biological biofilm-formation assays that fits with their structural work and proposed function roles, but the manuscript lacks solution-based, biophysical characterisation that might support the effects of the proline peptide bond and the Saf-Saf contacts i.e. NMR, SAXS, native MS for example.

Language needs tidying in places.

If these issues can be addressed, this referee would be more supportive especially as I believe this is from a relatively junior group.

[Editors' note: further revisions were requested prior to acceptance, as described below.]

Thank you for resubmitting your work entitled "Structural basis of host recognition and biofilm formation by *Salmonella* Saf pili" for further consideration at *eLife*. Your revised article has been favorably evaluated by Gisela Storz (Senior Editor) and a Reviewing Editor.

The manuscript has been improved but there are some remaining issues that need to be addressed before acceptance, as outlined below:

1) Recombinant Saf pili in *E. coli*.

The identity of the pili shown by EM as Saf pili is not convincing. The thickness and relative straightness and rigidity of the pili indicates they are helically wound rods, which is in contradiction with the thin fibrillary nature of Saf seen by Salih et al. and also seen for the related Afa fibers, as well as with the structural data on the SafA-SafA contact as shown in this report. It would be necessary to perform the experiment in a fim knockout background, to rule out the fibers shown are not type 1 pili. Since recombinant Saf are produced using the pASK-IBA4-Saf construct, can the fibers be stained with anti-strep immunogold?

Western analysis of SafA-Strep expression levels.

The blot shows similar expression levels of SafA, but it is not confirmed whether these are surface localized or periplasmic since the OM prep protocol is not clarified. Is it clear that the "outer membrane protein extraction kit (BestBio, China)" used selectively extracts OM fractions whilst not also retaining the soluble protein fraction? I could not find reference to the exact kit based on the provided information.

We also remain puzzled by the flow cytometry data. Why do the induced *E. coli* pASK-IBA4-Saf cells not show an altered FSC compared to the vector control? If the cells start to clump, that should be readily seen in the forward scatter profile.

2) SafDAA – SafDAA contact.

The SAXS experiments still are not convincing. The theoretical and observed scattering curves differ markedly. What is the chi-squared value for the fit? The authors mention "recombinant SafDAA-*dsc* might not be monodisperse in solution. Indeed, gel filtration characterization showed that SafDAA-*dsc* can undergo high-order oligomerization (Figure 6)." Monodispersity is a prerequisite to calculate meaningful scattering envelopes. The Guinier analysis of the SAXS data at different concentrations should inform on the dispersity of the analysed sample.

How do the authors explain that the SafDAA-SafDAA contact cannot be seen by SPR?

The cell aggregation data for the structure-based mutants (T7A/R9A/N94A and E8A/Q10A) are not in support of the claimed importance of the SafDAA-SafDAA contact seen in the crystals.

---

## [Author Response]

Essential revisions:1) The relative contribution / requirement of SafD needs to be established in the cell-based adhesion assay. This can be done by repeating the assay with a SafAA construct.

As suggested by reviewers, the cell adherence assay is now repeated with SafAA-*dsc* (Figure 4). Compared to SafDAA-*dsc* and SafDA-*dsc*, SafAA-*dsc* displayed little binding towards IPEC-J2 cells, suggesting that SafD is indeed required for the Saf-driven cell adherence. Furthermore, based on the observation that SafDAA-*dsc* showed stronger binding than that of SafDA-*dsc* and SafD*dsc*, we proposed a zipper-in mechanism for poly-SafAs (Figure 4).

2) The authors need to demonstrate recombinant Saf pili are expressed in their E. coli-based assays. The flow cytometry is flawed as done now because of the presence of the 50 mM EDTA. Saf pilus formation also needs confirmation by electron microscopy.

We appreciate reviewers’ concern. Firstly, the new flow cytometry experiment is repeated in the absence of EDTA (Figure 7). Secondly, the overexpression and surfacing of Saf pili and mutants are confirmed by electron microscopy and outer membrane extraction (Figure 7 and Figure 7—figure supplement 1).

3) The bead assay needs to be made quantitative. How representative are the current images? Why use a bead assay if the authors can look at bacterial clustering directly using light microscopy?I agree with reviewer 3 that a better alternative to the bead assay is solution-based affinity measurements for SafAAD self-association. By BLI, ITC or SPR and possibly supplemented with SAXS.

As suggested by reviewers, the bead assays of SafDAA-*dsc*, SafD-*dsc*, SafA-*dsc* and BSA (control) are now quantified and subjected to statistical analysis (Figure 5—figure supplement 1). Furthermore, light microcopy is used to monitor the bacterial clustering mediated by WT Saf pili and mutants. The representative visualization and quantitative analysis are shown in the revised Figure 7. Moreover, in order to demonstrate the presence of Saf-Saf self-association in solution, SAXS technique is used to characterize the SafDAA-*dsc* oligomerization. As shown in the revised Figure 5, SAXS analysis revealed an elongated envelope, supportive of crystallographic SafDAA-SafDAA dimer. More importantly, SEC-MALS characterization of T7A/R9A/N94A, E8A/Q10A and P20A showed that, when mutations were introduced to the crystal contact or the proline residue, the perturbation consistently disrupted the oligomerization activity of SafDAA in solution (Figure 6). All these results, together with the observations in cell aggregation and biofilm formation assays, support a biological relevant of Saf-Saf oligomerization in cell-cell interaction (Figure 9). However, we failed to measure the binding affinity of SafDAA-SafDAA using SPR technique. Consistently, at low concentration, the SafDAA-*dsc* displayed mainly as monomer as characterized by gel filtration (Figure 6). This, however not surprising, makes senses in the context of the biofilm development (because the tight and rigid intermolecular interaction might prevent the dynamic association and disassociation between bacterium and biofilm community during bacterial pathogenesis). In line with this observation, similar dynamic oligomerizations are also observed in other bacterial self-association such as Hap-Hap, Ag43-Ag43 etc.

4) If using the biofilm assay to compare point mutants, the authors need to evaluate whether the mutants do not alter Saf expression on the cell surface.

As suggested, the expression and surfacing of the mutants are quantified by outer membrane extraction and western blot analysis (Figure 7—figure supplement 1). The mutants showed similar expression levels as that of WT, suggesting that the structure-based mutations do not alter the expression/surfacing of Saf pili in *E. coli*. This result will help the interpretation of biofilm formation assay.

5) The authors need to address whether these observations are artifacts of crystallography. By definition, crystals are built using intermolecular contacts, including from hydrogen bonds and electrostatic interactions. Furthermore, conformational changes can be induced in order to propagate an ordered crystal. Are the contacts and observations artifacts or real?

We appreciate editor/reviewers’ concern. In this report, a variety of techniques including SAXS, SEC-MALS, bead/cell aggregation assay and biofilm formation assay (in which Saf/mutants expression are monitored by flow cytometry, EM and outer membrane extraction) are used to verify the crystallographic observation. In order to verify the crystallographic observation in crystal, the SAXS technique was to use characterize the recombinant SafDAA-*dsc* in solution. As shown in Figure 5, the molecular shape derived from SAXS scattering fits well with the crystallographic SafDAASafDAA dimer. Moreover, the structure-based mutations targeting the Type I and Type II dimerizations consistently impaired SafDAA-SafDAA self-association activity in SEC-MALS assay (Figure 6) further support the discovery of this report. More importantly, when the mutations are characterized in the context of full-length Saf pili in *E. coli*, similar disruptive results are observed (Figure 7). All these results have led to the proposal of physiological Saf-Saf association in bacterial aggregation, colony formation and ultimate biofilm formation.

6) The authors describe biological biofilm-formation assays that fits with their structural work and proposed function roles, but the manuscript lacks solution-based, biophysical characterisation that might support the effects of the proline peptide bond and the Saf-Saf contacts i.e. NMR, SAXS, native MS for example.

We thank reviewers’ kind and critical advice. As suggested, in order to verify the presence of SafDAA-SafDAA dimer in solution, SAXS characterization is used. The SAXS scattering suggests an elongated shape of SafDAA-*dsc* in solution. This molecular envelope significantly exceeds the length of SafDAA monomer, but fits well with the crystallographic dimer (Figure 5). However, as implied by the variation/differences in crystal fitting (Figure 5), the recombinant SafDAA-*dsc* might not be monodisperse in solution. Indeed, gel filtration characterization showed that SafDAA-*dsc* can undergo high-order oligomerization (Figure 6). In order to check the crystal contact and proline residue in the context of self-association, SEC-MALS technique is used. Consistent with SAXS characterization, WT SafDAA-*dsc* displays great tendency of high-order oligomerization/aggregation. In marked contrast, structure-based mutants significantly disrupt the self-associating activity (Figure 6). Also, see above, we also discuss the observation of SafDAA-*dsc* monomer in gel filtration in the context of other self-associating adhesins in the revised manuscript. In summary, these biophysical observations, together with other validations presented in this report, suggest that 1) the proline residue in Loop_G-A_ is an interesting structural determinant in pili assembly and function; 2) the Saf-Saf intermolecular oligomerization is physiological relevant, underpinning the process of bacterial aggregation and colony formation.

Below are the verbatim comments of the reviewers from which the above were distilled. We reproduce these here for your benefit although the most critical comments for your attention are those listed about under "essential".Reviewer #1:[…] 1) There are many grammatical errors throughout the manuscript that should be corrected before publication.

As suggested, the grammatical and typo errors are corrected in the revised manuscript.

2) The authors suggest that the SafD binding site is located at an electronegative patch on the side of the SafD molecule, but fail to discuss this further. These residues could be highlighted on Figure 2E to allow for comparison to the known binding residues of other members of the AfaD family. There was no attempt to mutate any of the residues making up this surface to probe their role in binding to IPEC-J2 cells assessed.

As suggested, the putative SafD binding site is subjected to mutagenesis and cell adherence assay. Thus far, single point mutations failed to abolish the interaction between SafDAA-*dsc* and IPEC-J2 cells (see Author response image 1). Although the results are less supportive, we still want to draw reader/colleagues’ attention to this interesting structural feature in SafD. In light of the observation that AfaD (a SafD homolog) recognize host integrin (possibly via electrostatic interaction), it is reasonable to suggest a putative binding site for more vigorous study in the future.

**Author response image 1. respfig1:** Preliminary characterization of the putative host binding site in SafD.

3) The increase in binding affinity of the SafDAA construct vs, the SafDA construct could either mean that the second SafA subunit interacts with the host receptor directly, or that its presence results in subtle conformation changes that alter the affinity of the two terminal Saf subunits for its ligand. The superposition of the two SafD structures is found in Figure 2D; however, this analysis is limited to the conformation of Tyr13. Are there other structural changes in SafD? Similarly, are there any SafA side chains whose positions differ significantly in the I and L forms, either within each conformation or between the two conformations that may indicate structural changes underlying the affinity difference?

In order to address the contribution of SafA in cell-host interaction, we have repeated the cell adherence assay using SafAA-*dsc* (also see our reply to Essential revisions 1). As shown in Figure 4, the current data suggest a secondary binding role of SafA, in which the tip complex of SafD-SafA1 is required for cell adherence and the poly-SafA might contribute to host recognition in the subsequent steps to facilitate more intimate interactions (Figure 4). Furthermore, as suggested by reviewer, the residues, Glu8 and Gln10 engaging inter-molecular interaction are compared (Figure 3—figure supplement 1). In particular, Glu8 and the adjacent loop showed conformational change upon dimerization, suggestive of functional role in Saf-Saf interaction. This is in line with our structure-based mutagenesis analysis presented in Figure 6 and 7.

4) The I/ for the outer resolution shell of the SafDAA-dsc structure is only 0.8. Without seeing the density or knowing the CC1/2 in this shell, a resolution cutoff of 2.6 Å may not be appropriate. The B-factors and high R_free_ in the outer shell are also concerning.

As suggested, the information of CC1/2 is updated in the revised Table 1. According to CC1/2 and electron density map (see Author response image 2), a new resolution cutoff of 2.8 Å for SafDAA*dsc* is used.

**Author response image 2. respfig2:** Electron density of SafDAA-*dsc*.

Reviewer #2:[…] Abstract:"Although Saf pilus is ubiquitously expressed in Salmonella enterica, its structure and function remain poorly understood" – this is a strong claim without reference, are Saf pili ubiquitously expressed? Presence in the genome does not equal expression. In general, the available literature on Saf biology and previous functional studies on fimbriae in Salmonella pathogenesis need to be better described in the introduction. The claim is repeated in the Introduction, again without any reference.

We have rephrased the sentence. The new text is “Although Saf pilus is frequently observed in *Salmonella enterica*, its structure and function remain poorly understood”.

"More importantly, Saf-Saf oligomerization helps to define a universal, inter-cellular self-associating mechanism in bacterial aggregation/colonization." What is the universal self-association mechanism that is proposed?

As suggested by reviewer, the text explaining the universal self-association mechanism is included in the revised manuscript, “More importantly, Saf-Saf structures and functional characterizations help to define a pili-mediated inter-cellular oligomerization mechanism for bacterial aggregation, colonization and ultimate biofilm formation”.

Introduction:"Although Saf pili are ubiquitously expressed in Salmonella enterica, it is not clear how Saf pili mediate host recognition, and whether they participate in cell-cell interaction and biofilm development. In this study, we aim to provide more structural and functional insights into this important pilus subtype." Any claims on the role or importance of Saf pili in Salmonella need references or new experimental data.

As suggested, the sentence is rephrased to make clear to readers that Saf genes are frequently observed in *Salmonella enterica*.

Results:"Furthermore, based on sequence alignment (Figure 2 and Figure 1—figure supplement 1), SafD is thought to be adhesive." This is a non-logical claim as currently presented. If referring to sequence homology with E. coli AfaD, is it expected that SafD acts as an invasin? This should then be specified and included in the Discussion.

Current data do not support an invasin role of SafD. However, we agree with reviewer’s suggestion that SafD share significant homology (30%) with *E. coli* AfaD, and hence might function as an invasin. In the revised manuscript, we have made this information clear to readers in results and Discussion sections.

"However, in P* position (a favored interaction that catalyze/stabilizes the exchanged product during DSE), Fhe3 is no longer in direct interaction with SafD." "Fhe3" should be corrected to "Phe3" throughout manuscript. What is the "catalytic" role the authors are referring to?

We thank reviewer’s critical reading. The typo error is now corrected. Furthermore, in order to avoid confusion, the text of “catalyze/stabilize” is now replaced with “stabilize”.

"The superimposition of the SafD molecules shows interesting structural variations/flexibilities in LoopA-B on top of the P5 pocket, implying this loop might play a regulatory role in DSE (Figure 2)." Why does the presence of two conformations in the LoopA-B imply a regulatory role? What kind of regulatory role do the authors have in mind and evidence is there to invoke such role?

We understand reviewer’s concern and demand of experimental support for this claim. Thus far, the hypothesis of a regulatory role of the Loop_A-B_ over the P5 pocket mainly stems from the structural observation (Figure 2). Consistently, the flexibility of Loop_A-B_ is frequently observed in other pilin subunits (Figure 2—figure supplement 1). In addition, MD simulation of *E. coli* pilin-Nte complex suggested a regulatory of Loop_A-B_ in P-pili (Ford *et al.,* 2012). Therefore, by comparison, the Loop_A-B_ in Saf might share a similar structural role during pili assembly. However, we appreciate reviewer’s concern, and have made it clear to readers in the revised manuscript that the claim is “*speculative*” and further experimental data is required to verify the regulatory role of Loop_A-B_ in Saf.

"The SafD, SafA1, SafA2 and SafA3Nte have assembled into a thread-like shape with 26 Å in diameter and 136 Å in length/height (Figure 3 and Figure 3—figure supplement 1)." Why are linker sequences not shown and are N- and C-termini labelled in the Saf subunits? In the supplementary figure the linkers are shown.

Thanks for reviewer’s critical reading. We now include the DNKQ, in between SafDSafA1, in the revised Figure 3. In addition, the N- and C-termini are labeled for all subunits to allow readers to follow the assembly of SafD-SafA1-SafA2-SafA3_Nte_.

"In between SafD and SafA1, Pro20 adopts a cis-configuration, enabling a kink in the subunit-subunit linker loop that, in turn, allows the formation of intra-molecular hydrogen bonds (Figure 3—figure supplement 1)." Do the authors mean "intermolecular"? Also, what is shown in Figure 3—figure supplement 1? The labelled residues do not make sense. T111 looks like an Ile, or did the authors include the -OH hydrogen? Q134 and N6 do not make sense as Q or N, or again were the -NH2 hydrogens included? If including hydrogens, that should be done for all residues and all atoms in a residue; or clearly specified in the legend. If they are hydrogens, the Q134 side chain amide needs a 180 degree rotation.

We thank reviewer’s critical reading. The display errors are now corrected.

"Collectively, these data have suggested a poly-adhesive activity of Saf pili, in which SafD and SafA subunits can bind concertedly to the unknown host cell receptors to enable intimate host:bacterium interaction and colonization via a zip-in mechanism (Figure 4)." What is the contribution of SafD in the polyadhesive binding? Does a SafA polymer or SafAA bind the IPEC-J2 cells?

SafAA is now included in the new cell adherence assay (Figure 4). Please also see our reply to editor (Essential revisions).

"As shown in Figure 5, the beads coated with the recombinant SafDAA-dsc protein aggregated together to form clusters." The authors use a bead agglutination assay to evaluate the self-association of SafDAA-dsc. The experiments as shown are not conclusive. How representative are the images? A more quantitative assay or representation of the bead assay is required to draw conclusions.

Visualization and quantitative analysis of bead/cell aggregation assays are shown in Figure 5—figure supplement 1 and Figure 7, respectively. Please also see our reply to editor (Essential revisions).

For the E. coli cell sorting and biofilm assay (Figure 6), it is not currently demonstrated that surface-exposed Saf pili are indeed expressed (see below comments in the Materials and methods) and for the evaluation of the mutants (Figure 6), how do the authors differentiate whether reduced biofilm titers stem from reduced self-association versus reduced Saf pilus expression? If surface-exposed Saf pili are formed, are the amounts obtained from recombinant plasmid-based expression representative for those seen under native conditions in Salmonella? Is the N-terminal Strep tag on SafA in the recombinant saf operon compatible with Saf pilus formation?

Please also see our reply to editor (Essential revisions). In brief, the expression of Saf pili and mutants are monitored by flow cytometry, EM and outer membrane extraction (Figure 7 and Figure 7—figure supplement 1). Furthermore, EM characterization shows that the presence of strep tag does not interfere with the expression/surfacing of Saf pili (Figure 7—figure supplement 1).

"At the initial stage, Saf pili might interact with each via Type I dimerization. As biofilm develops, the Saf pili can proceed into Type II dimerization, by each step of which the two adjacent cells move ~40 Å towards each other, giving rise to further intimate association and aggregation, leading to the ultimate micro-colony and biofilm formation (Figure 8)." It is not clear what the sequential model is based on. Why would type I interactions take priority on Type II interactions?

We agree with reviewer’s comment that the sequential model of biofilm formation might be overreaching at this stage. The revised model is proposed, in which Type I and Type II Saf-Saf oligomerizations could contribute to cell-cell interaction, bacterial aggregation and micro-colony formation (Figure 9).

Discussion:"The movement of LoopA-B and Try13 sidechain appears to act as a molecular "switch", allowing or precluding the access of incoming Nte peptide" There is no experimental basis for this claim – see also comment above.

See our reply to above. We have made it clear to readers that the claim is “speculative” and demands further experimental support.

"It is reasonable to believe that the further variation of Pro20 isomerization and torsion angles will no doubt allow much versatile architecture of Saf pili that is necessary for the adherence and survival in different surfaces and environments." It is not clear what is meant here. Pro will be cis or trans, no further variation, and the dihedral angles of X-Pro-X bounds have reduced conformational freedom compared to that of other residues. Also what is meant with the concluding remark on the previously unrecognized role of proline in the G-A loop is unclear.

We have revised the text to make it clear to readers that 1) proline is a special amino acid, the isomerization of which is influential protein folding and overall assembly; 2) Unexpectedly, the structural finding in this report shows that the dihedral angles of X-Pro20-X might be an important structure determinant shaping the SafDAA assembly into an “I”- or “L”-like architecture; 3) More importantly, mutations of Pro20 results in reduced Saf-Saf oligomerization, cell-cell aggregation and biofilm formation (Figure 6 and Figure 7); 4) the proline residue is frequently observed in LoopG-A connecting subunits in pili assembly. As shown in Figure 8, different isomerizations and torsion angles of X-Pro-X (X for any adjacent amino acid) is instrumental in overall pili assembly and architecture. In order to highlight this observation to readers (but not over-stretching the interpretation), we have rephrased the claim in, “In summary, the SafDAA-dsc structure, together with previous observations (Figure 8), might highlight a structural/functional role of proline residue in Loop_G-A_, by the movement of which bacterial pili can assemble into diversive morphologies for adaptation of different environments (Hung et al., 1996; Zav'yalov et al., 2010)”.

"The cell adherence assay presented in this report showed that both SafA and SafD subunits are important for the binding of host cell receptors." The current data do not proof the involvement of SafD in the adherence – see comment above. Also the claim "The mini Saf pilus, SafDAA-dsc, appeared to mimic the role of Saf pili" should be reworded. There is no data on the Saf pilus presented, or referred to in the Introduction.

Please also see our reply to editor (Essential revisions). We thank reviewer’s kind suggestion of SafAA-*dsc* characterization in the cell adherence assay, which has improved the understanding/interpretation of Saf-mediated host recognition.

"Based on sequence alignment, the host receptors integrin beta3 and fibronectin of AfaD (~30% sequence indentity with SafD) might be possible targets for Saf pili that should be further investigated in future studies (Cota et al., 2006)." The reference shows AfaD binding to integrin beta1. Also, there is several studies showing that pilus adhesins with seq. ID's well above 50% percent can have unrelated receptors. Is there evidence that SafD acts as invasion similar to E. coli AfaD?

See our reply above. In the revised manuscript, we clearly state that the host receptor is not yet identified. However, the current knowledge of AfaD and other SafD homologs is discussed with current observation in SafD adhesin. As for SafD as invasin, it is purely speculative, solely based on sequence alignment with AfaD (Results).

Materials and methods:Flow cytometry analysis is used to evaluate the surface expression of Saf pili on E. coli. The Materials and methods describe use of 50mM EDTA for fixing and antibody based staining of the cells. EDTA treatment permeabilizes the outer membrane. How can the authors ensure that surface-localized Saf pili were formed? Also, if Saf pili induce self-aggregation, is the flow-cytometry experiment looking at single cells or cell clusters? This should be apparent from the scattering levels.

Please also see our reply to editor (Essential revisions). In brief, the new flow cytometry assay is performed in the absence of EDTA. In addition, EM and outer membrane extraction together western blot analysis are used to confirm the levels of Saf expression and surfacing in *E. coli* (Figure 7 and Figure 7—figure supplement 1). In order to ensure the analysis of single cells (not cell clusters) in flow cytometry, 2 µm fluorescent blue polystyrene latex beads were used as a search temple. The scattering levels are shown in Figure 7.

Reviewer #3: […] One major question the authors need to address is whether these observations are artefacts of crystallography. By definition, crystals are built using intermolecular contacts, including from hydrogen bonds and electrostatic interactions. Furthermore, conformational changes can be induced in order to propagate an ordered crystal. Are their contacts and observations artefacts or real?

Please see our reply to editor (Essential revisions). In the revised manuscript, we have used SAXS, SEC-MALS, bead/cell aggregation assay, biofilm formation assay to analyze/verify our finding in X-ray crystallography. All the evidences support a physiological relevant Saf-Saf oligomerization in bacterial aggregation.

The authors describe biological biofilm-formation assays that fits with their structural work and proposed function roles, but the manuscript lacks solution-based, biophysical characterisation that might support the effects of the proline peptide bond and the Saf-Saf contacts i.e. NMR, SAXS, native MS for example.

We thank reviewer’s kind suggestion. Please see our reply to editor (Essential revisions). We have included biophysical technique such as SAXS and SEC-MALS to characterize the SafDAA*dsc* dimers in solution. These results are in line with the bead/cell aggregation and biofilm formation assays, supportive of the major finding of this report.

Language needs tidying in places.

We have tried our best to tidy up the writing in the revised manuscript.

[Editors' note: further revisions were requested prior to acceptance, as described below.]

The manuscript has been improved but there are some remaining issues that need to be addressed before acceptance, as outlined below:1) Recombinant Saf pili in E. coli.The identity of the pili shown by EM as Saf pili is not convincing. The thickness and relative straightness and rigidity of the pili indicates they are helically wound rods, which is in contradiction with the thin fibrillary nature of Saf seen by Salih et al. and also seen for the related Afa fibers, as well as with the structural data on the SafA-SafA contact as shown in this report. It would be necessary to perform the experiment in a fim knockout background, to rule out the fibers shown are not type 1 pili. Since recombinant Saf are produced using the pASK-IBA4-Saf construct, can the fibers be stained with anti-strep immunogold?

We really appreciate editor/reviewers’ critical comments. In the revised Figure 7, anti-strep immunogold staining and EM visualization were used to monitor the surfacing of Saf pili. This, together with the results observed in the experiments of flow cytometry and outer membrane extraction (Figure 7), should give sufficient evidences to confirm the expression/secretion of Saf pili in *E. coli*.

Western analysis of SafA-Strep expression levels.The blot shows similar expression levels of SafA, but it is not confirmed whether these are surface localized or periplasmic since the OM prep protocol is not clarified. Is it clear that the "outer membrane protein extraction kit (BestBio, China)" used selectively extracts OM fractions whilst not also retaining the soluble protein fraction? I could not find reference to the exact kit based on the provided information.

In the revised manuscript, the protocol and references describing the preparation of outer membrane fractions are detailed in the Materials and methods section. Besides, in order to validate the feasibility and selectivity of western analysis, a negative control pASK-IBA4-SafABD (i.e. the ∆SafC mutation that was designed to abolish the surfacing/assembly of Saf pili in the outer membrane) was included to estimate the expression levels of WT Saf pili and mutants. (Figure 7).

We also remain puzzled by the flow cytometry data. Why do the induced E. coli pASK-IBA4-Saf cells not show an altered FSC compared to the vector control? If the cells start to clump, that should be readily seen in the forward scatter profile.

Thanks to reviewers’ suggestion, we have revisited the flow cytometry experiment using DMEM, a medium often used in the studies of bacterial aggregation and biofilm formation. As reported before (MacKenzie et al., 2010; Torrent et al., 2012), FSC signal derived from flow cytometry analysis could be used to monitor the size of the object, and hence cell aggregation. In order to check the biofilm formation activity, the cells of pASK-IBA4-Saf were grown in the minimum medium DMEM before they were subjected to the analysis of flow cytometry. As shown in the revised Figure 7—figure supplement 1, the cells expressing Saf displayed stronger preference to undergo cell-cell aggregation than control cells. The FCS signal showed that 13.8% Saf-expressing cells clumped together. In control cells (i.e. without Saf pili), only ~1.9% aggregated. This is in good agreement with the observation reported in cell aggregation and biofilm formation assays (Figure 7), supportive of an important role of Saf pili in bacterial aggregation.

2) SafDAA – SafDAA contact.The SAXS experiments still are not convincing. The theoretical and observed scattering curves differ markedly. What is the chi-squared value for the fit? The authors mention "recombinant SafDAA-dsc might not be monodisperse in solution. Indeed, gel filtration characterization showed that SafDAA-dsc can undergo high-order oligomerization (Figure 6)." Monodispersity is a prerequisite to calculate meaningful scattering envelopes. The Guinier analysis of the SAXS data at different concentrations should inform on the dispersity of the analysed sample.

As suggested by reviewers, we have performed more SAXS characterizations of SafDAA-*dsc* at different concentrations (i.e. 1, 2.5 and 5 mg/ml). As shown below, the Guinier analysis of these data showed that the samples were mono-dispersive and suitable for SAXS analysis (Figure below, 1A). In addition, the Kraky plots of distinct peaks suggested that SafDAA-*dsc* molecules might have multiple oligomerization stages in solution (Figure below, 1B). This has led to the OLIGOMER analysis by program CRYSOL. Under this approach, the SafDAA-*dsc* monomers, dimers and trimer (observed in crystallography) were all used to fit the experimental SAXS data. The result are shown in the revised Figure 5EF and the Figure below. Compared to the previous fitting, the Chi^2^ values has dropped significantly to the level of 2.74 (for 1 mg/ml sample), suggesting a dynamic multimeric assembly in solution containing monomer (~79.7%), dimer (~2.1%) and trimer (~18.2%) (Figure 5).

**Author response image 3. respfig3:** SAXS characterization of SafDAA-*dsc* at concentrations of 1, 2.5 and 5 mg/ml. (**A**) The Guinier analysis. (**B**) The Kraky analysis. (**C**) The fitting between the SAXS experimental data and crystallographic structures of SafDAA-*dsc* monomer/dimer/trimer. The fitting result of SafDAA-*dsc* (1 mg/ml) was shown in the revised Figure 5.

How do the authors explain that the SafDAA-SafDAA contact cannot be seen by SPR?

The failure of detecting SafDAA-SafDAA interaction by SPR indicates that the nature of Saf-mediated association might be weak. However, this is not surprising. Similar weak interacting nature were also reported in other inter-cellular cross-linkers such as *H.influenzae* Hap, *E. coli* Ag43 etc. (Meng et al., 2011; Heras et al., 2014). In previous study, it has been proposed that the polymerization of inter-molecular Haps and Ag43s might be a formidable force for cell-cell interaction. Under the same principle, we could envisage a similar approach by Saf pili: the progressive inter-cellular oligomerization between multiple Saf polymers might be sufficient to trigger bacterial aggregation (as demonstrated in Figure 7 and Figure 7—figure supplement 1). For now, based on the data in hand, we could speculate that the weak SafDAA-SafDAA might be biological relevant and a necessary structural determinant/factor, enabling the polymerization/depolymerization of Saf pili and the association/disassociation between single cell and biofilm community, in turn regulating the bacterial life cycles of staying with other cell (to seek protection), or leaving from the cell community (to seek further colonization and infection). Again, this is purely speculative and should demand more investigation in the future.

The cell aggregation data for the structure-based mutants (T7A/R9A/N94A and E8A/Q10A) are not in support of the claimed importance of the SafDAA-SafDAA contact seen in the crystals.

We understand reviewers’ concern, and have acknowledged the weak interacting nature of SafDAA-SafDAA in the revised manuscript. Supported by the new SAXS analysis, the SafDAA-SafDAA multimers revealed by crystallography seem plausible in solution. In addition, the SEC-MALS characterization of WT SafDAA and mutants showed that the mutation targeting the SafDAA-SafDAA contact arrested the SafDAA interaction/oligomerization in a dimeric stage. This, together with SAXS data, supports the idea of self-associating activity of SafDAA-*dsc*. More important, flow cytometry, beads/cells aggregation assays and biofilm formation assays all in good agreement with each other, supporting the self-associating activity of SafDAA-*dsc* and Saf-Saf interaction in bacterial aggregation. This has not been reported before. However, we appreciate reviewers’ concern that, in cell-aggregation assay, the structure-based mutants still formed colonies, but with less quantity and different morphology. Interestingly, similar mutants also displayed less biofilm activity, suggestive of certain degree of disruptive impact on Saf-Saf contact and bacterial aggregation. Based on these observations and the results in biophysical characterizations, we have rephrased the discussion and make clear to readers that Saf pili might adopt multiple configurations in inter-molecular interactions that will include, but not restricted to, the intertwining/oligomerization unveiled by the crystal structures of SafDAA-*dsc* multimers.